# Timing and origin of natural gas accumulation in the Siljan impact structure, Sweden

Henrik Drake [1]*, Nick M.W. Roberts [2], Christine Heim [3], Martin J. Whitehouse [4], Sandra Siljeström[5], Ellen Kooijman[4], Curt Broman[6], Magnus Ivarsson[4,7] & Mats E. Åström[1]

Fractured rocks of impact craters may be suitable hosts for deep microbial communities on Earth and potentially other terrestrial planets, yet direct evidence remains elusive. Here, we present a study of the largest crater of Europe, the Devonian Siljan structure, showing that impact structures can be important unexplored hosts for long-term deep microbial activity. Secondary carbonate minerals dated to 80 ± 5 to 22 ± 3 million years, and thus postdating the impact by more than 300 million years, have isotopic signatures revealing both microbial methanogenesis and anaerobic oxidation of methane in the bedrock. Hydrocarbons mobilized from matured shale source rocks were utilized by subsurface microorganisms, leading to accumulation of microbial methane mixed with a thermogenic and possibly a minor abiotic gas fraction beneath a sedimentary cap rock at the crater rim. These new insights into crater hosted gas accumulation and microbial activity have implications for understanding the astrobiological consequences of impacts.

[1] Linnæus University, Department of Biology and Environmental Science, 39182 Kalmar, Sweden. [2] Geochronology and Tracers Facility, British Geological Survey, Nottingham NG12 5GG, UK. [3] Department of Geobiology, Geoscience Centre Göttingen of the Georg-August University, Goldschmidtstr. 3, 37077 Göttingen, Germany. [4] Swedish Museum of Natural History, P.O. Box 50 007, 10405 Stockholm, Sweden. [5] Bioscience and Materials/Chemistry and Materials, RISE Research Institutes of Sweden, Box 5607, 114 86 Stockholm, Sweden. [6] Department of Geological Sciences, Stockholm University, 106 91 Stockholm, Sweden. [7] Department of Biology, University of Southern Denmark, Campusvej 55, 5230 Odense, Denmark. *email: henrik.drake@lnu.se

mpact craters and associated impact-generated hydrothermal systems may be favorable for microbial colonization on Earth and potentially other planets[1,2]. Extensive fracturing at depth caused by impacts provides pore space, and heat generated by the impact drives hydrothermal convection, favorable for deep ecosystems[3,4]. Although very few studies of the deep biosphere in impact structures exist, a small number of reports of fossil- and geochemical signatures support post-impact colonization of impact hydrothermal systems[5,6]. However, direct temporal constraints of mineralized remains of microorganisms in rock fractures are generally lacking, and the link between the impact and subsequent colonization thus remains elusive. Confirmation of impact craters as favorable environments for deep biosphere communities would substantially enhance our present understanding on deep energy cycling of these systems and involve considerable astrobiological implications[3,7]. Investigation of methanogens and methanotrophs are of special interest since methane emissions, both seasonal and as isolated spikes, have recently been detected at Martian craters[8].

In the largest impact structure in Europe, the Late Devonian (380.9 ± 4.6 Ma[9]) Siljan crater in Sweden, the past and present deep biosphere communities remain unexplored, but the structure has been thoroughly studied for potential methane accumulation. This strong greenhouse gas can form via three main mechanisms in the deep subsurface: abiotic, that is, during inorganic reactions of compounds, e.g. $H_2$ and $CO_2$; thermogenic, that is, by organic matter breakdown at high temperatures; and microbial activity[10,11], which has been largely overlooked at Siljan and other impact structures.

In the late 1970s and 1980s, astrophysicist Thomas Gold put forward controversial theories of mantle-derived methane ascending through fractures to shallow crustal levels where it would accumulate and form higher hydrocarbons and petroleum[12,13]. Gold proposed that significant amounts of methane of mantle origin had ascended the impact-deformed basement at Siljan[14], and accumulated beneath a cap rock of carbonate-sealed fractures in the upper crust. Accordingly, from the late 1980s to early 1990s, deep exploratory wells were drilled in the central plateau of exhumed Paleoproterozoic granite[15], but no economic gas quantities could be established and the project was abandoned. The origin of the hydrocarbons found during the deep drillings remains disputed, not the least due to potential contamination from drilling lubricants[16]. Gold's theory is now considered invalid and has been overtaken by newer models on deep hydrocarbon formation[17]. Abiotic methane does occur in a variety of geological settings[10], including Precambrian shields, but the presence of a globally significant abiogenic source of hydrocarbons has generally been ruled out[18]. Recent studies of fractured Precambrian crystalline rocks have revealed deep methane occurrences of various, often complex origin, including microbial, and abiotic[19]. High methane concentrations in crystalline rocks are commonly associated with serpentinized ultramafic and graphite-bearing rocks[10,19], but at Siljan these rock types are not present and contribution of abiotic methane to the crystalline and sedimentary rock aquifers is yet to be proven.

In recent years, prospecting for methane has been re-initiated at Siljan by the prospecting company AB Igrene. This time the focus is on the fractured crystalline bedrock beneath 200–600 m thick[20] down-faulted Ordovician and Silurian sedimentary rocks (dominantly limestone but shales are also abundant) in the ring-like crater depression, where several cored boreholes have been drilled to 400–700 m depth (Fig. 1)[14]. Methane accumulations have been detected during the drilling campaigns, both in the sedimentary rock (proposed cap rocks) and deep within the granite fracture system, but no qualified estimate of total gas volumes has yet been made public.

Occurrences of seep oil associated with the Siljan crater sedimentary successions have, in fact, been known for hundreds of years, dating back to reports by Linnaeus in 1734[in21]. Seep oil and bitumen in limestone have been interpreted to have been generated from organically rich Upper Ordovician black shale[22,23]. The thermal maturity of this organic-rich shale as well as of Lower Silurian shale has reached the initial stage of oil generation, and hydrocarbons have migrated from these more mature sediments into marginally mature sediments[24]. The overburden and the time of burial apparently were sufficient to mature the potential source rocks at Siljan, although it has been speculated that heat effect of the meteorite impact locally matured the source rock instantaneously[23]. Still, the potential input of thermogenic gas to the deep granite aquifer at Siljan remains elusive.

The potential contribution of microbial methane at Siljan has, in contrast, largely been overlooked, and, consequently, the deep microbial communities are yet unexplored, which is the case for most terrestrial impact structures. Isotopic and biomarker clues to ancient microbial processes such as methanogenesis and anaerobic oxidation of methane (AOM) with associated sulfate reduction can be preserved within minerals formed in response to these microbial processes. These signatures can remain within the minerals over considerable geological time[25]. Relatively light, $^{12}C$-rich, methane is produced during microbial methanogenesis and, consequently, $^{13}C$ accumulates in the residual $CO_2$[26]. When these distinguished carbon pools are subsequently incorporated in secondary carbonate minerals the isotopic compositions are preserved such that $^{13}C$-enrichment marks methanogenesis[27], and $^{12}C$-enrichment AOM[28,29]. Advances in high spatial resolution U-Pb geochronology make it possible to gain timing constraints about discrete events of mineral precipitation following methane production and consumption in fractured rock[25,30].

Here, we apply both the carbon and U-Pb isotopic approaches in combination with analysis of organic compounds and gases to disclose accumulation of a mixed, but dominantly microbial, origin of methane of Cretaceous or younger age at Siljan. The isotopic mineral-gas dataset is the most comprehensive yet reported from any impact structure and provides new constraints for the unexplored deep microbial ecosystems of terrestrial impact craters, particularly regarding deep methane formation and consumption. Implications include both a broader perspective regarding natural gas accumulations in the upper crystalline crust, and the potential and challenges in understanding the significance of impacts as oases for life on otherwise dead planetary bodies.

## Results

**Stable carbon isotope composition of calcite.** Calcite that occurs together with sulfides and bitumen in secondary mineral coatings of open fractures (Figs. 2 and 3) shows a large $\delta^{13}C$ variability, in total 73.8‰ V-PDB ($n = 984$, Supplementary Data 1). The values are ranging from significantly $^{13}C$-depleted (−52.3‰) to $^{13}C$-enriched (+21.5‰). Many of the $\delta^{13}C$ excursions occur where extensive gas accumulations were observed during drilling (Fig. 4). The $\delta^{13}C_{calcite}$ range in the sedimentary rock fractures is −12.5‰ to +21.5‰ and in granite −52.3‰ to +18.9‰. Strongly positive $\delta^{13}C_{calcite}$ values occur in 38% of the fractures in the sedimentary rock and 28% in the granite, and are found up to 212 m above the sediment-granite contact (177 m below ground surface), as well as up to 214 m below the contact, to maximum depths of 620 m. The most $^{13}C$-depleted calcite (−52.3‰) is from the granite-sedimentary rock contact (Fig. 4a, b).

Samples with large $^{13}C$ depletion ($\delta^{13}C < -35‰$) or $^{13}C$-enrichment ($\delta^{13}C > +5‰$) either show relatively homogeneous

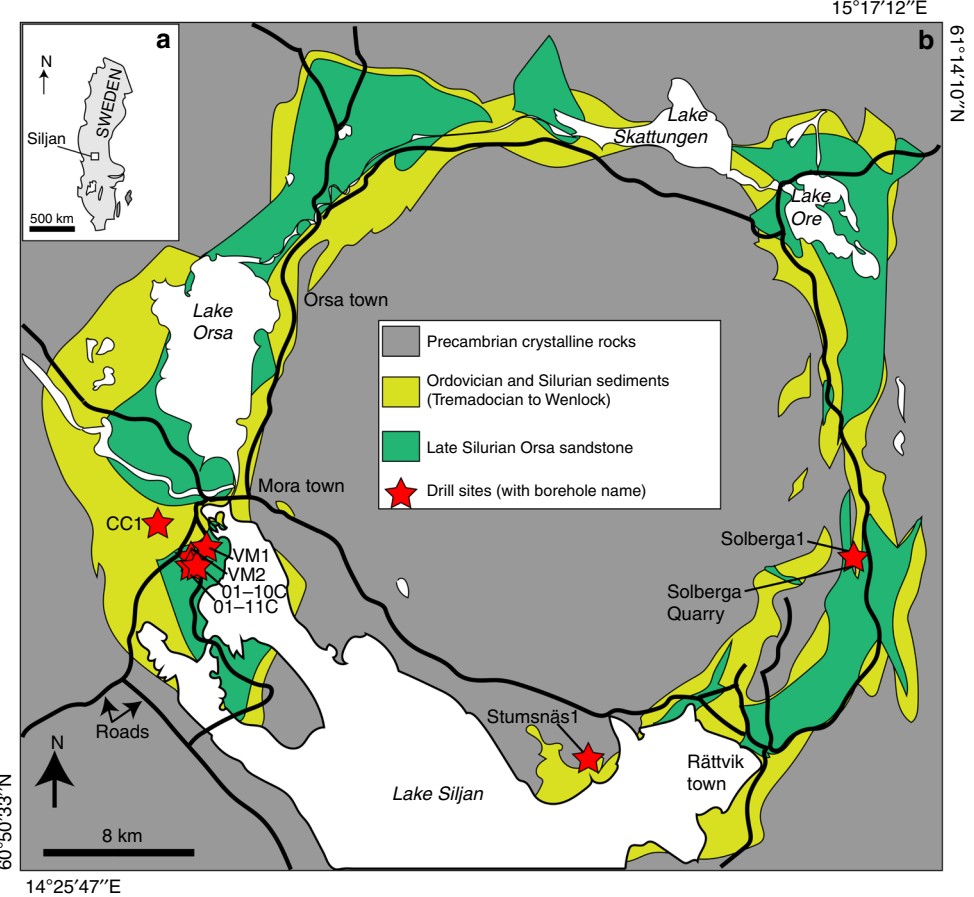

**Fig. 1** Maps of the Siljan impact structure and study locations. **a** Map of Sweden with the Siljan area indicated. **b** Geological map of the Siljan impact structure with locations of the cored boreholes and the quarry sampled for mineral coatings indicated, along with the sedimentary units in the crater depression, towns, lakes (white) and roads (black lines). Gas compositions exist from boreholes VM2 and VM5 (located adjacent to VM2). Modified from ref. [68]

composition throughout the crystals (Fig. 2o, r), or $^{13}$C enrichment or depletion at the outermost growth zones (Fig. 2p, q). The latter feature is accompanied by a significant increase in $^{87}$Sr/$^{86}$Sr values in the outermost growth zone compared to the inner part (Fig. 2q), in contrast to the relatively homogeneous $^{87}$Sr/$^{86}$Sr values of calcite with homogeneous $\delta^{13}$C values (Fig. 2r). Calcite in limestone fractures with abundant solid bitumen has highly positive $\delta^{13}$C values.

**Stable sulfur isotope composition of pyrite**. The $\delta^{34}S_{pyrite}$ values in fractured granite span 119.8‰, from −41.9‰ to +78.0‰ (Supplementary Data 2, $n = 443$). In sedimentary rock fractures the values are from −5.8‰ to +41.8‰. The $\delta^{34}S_{pyrite}$ value distribution within and between different crystals in the same fractures shows large variability. Notable examples include relatively homogeneous light $\delta^{34}S_{pyrite}$ values of −40 ± 1‰ (Supplementary Fig. 1b), increasing $\delta^{34}S_{pyrite}$ values from core to rim (Supplementary Fig. 1c), and small variation within individual crystals but substantial variation (93.7‰) between different crystals (Supplementary Fig. 1d).

**Fluid inclusions**. Only one of ten calcite samples examined contained fluid inclusions (Supplementary Data 3). These inclusions are of one and two-phase type. The latter have homogenization temperatures of 40–55 °C and ice melting temperatures equivalent to salinities of 1.6–2.7 mass % NaCl

($n = 6$). The general lack of fluid inclusions and the nature of the few detected inclusions indicate low-temperature origin.

**U-Pb geochronology**. Seven calcite samples gave U-Pb age solutions from the high spatial resolution analyses. $^{13}$C-enriched calcite from a limestone fracture at 212 m depth gave a single event age of 22.2 ± 2.5 Ma (Fig. 5b) whereas the $^{13}$C-enriched calcite at 170 m is more complex, with two isochrons; at 80 ± 5 Ma and 39 ± 3 Ma (Fig. 5a). In the granitic basement, ages of 39.2 ± 1.4 Ma and 65 ± 10 Ma were obtained when targeting the $^{13}$C-rich outermost calcite growth zones at 537 and 442 m depth, respectively (Fig. 5c, d). $^{13}$C-depleted calcite at the sediment-granite contact gave a 37.7 ± 1.9 Ma age but also an uncertain population at 464 ± 60 Ma ($n_{spots} = 3$, Fig. 5e). Calcite without any significant excursions in $\delta^{13}C_{calcite}$ values (−3.9 to +0.8‰) from two granite fracture samples yielded 506 ± 25 and 576 ± 64 Ma ages (Supplementary Fig. 2, full data and analytical details in Supplementary Data 4–6).

**Organic remains in the mineral coatings**. The calcite coatings analyzed for preserved organic compounds using gas-chromatography mass spectrometry (GC-MS, $n = 6$) represented $^{13}$C-rich calcite in limestone (VM2:212) and granite (CC1:539 and 608, VM1:442), as well as $^{13}$C-depleted calcite from the granite-sedimentary rock contact (VM1:251). Although overall low in organic content, the sample from limestone showed a clear unimodal distribution of n-alkanes ranging from n-$C_{17}$ to n-$C_{42}$ with a maximum at n-$C_{23}$ (Fig. 6a). The hydrocarbon range

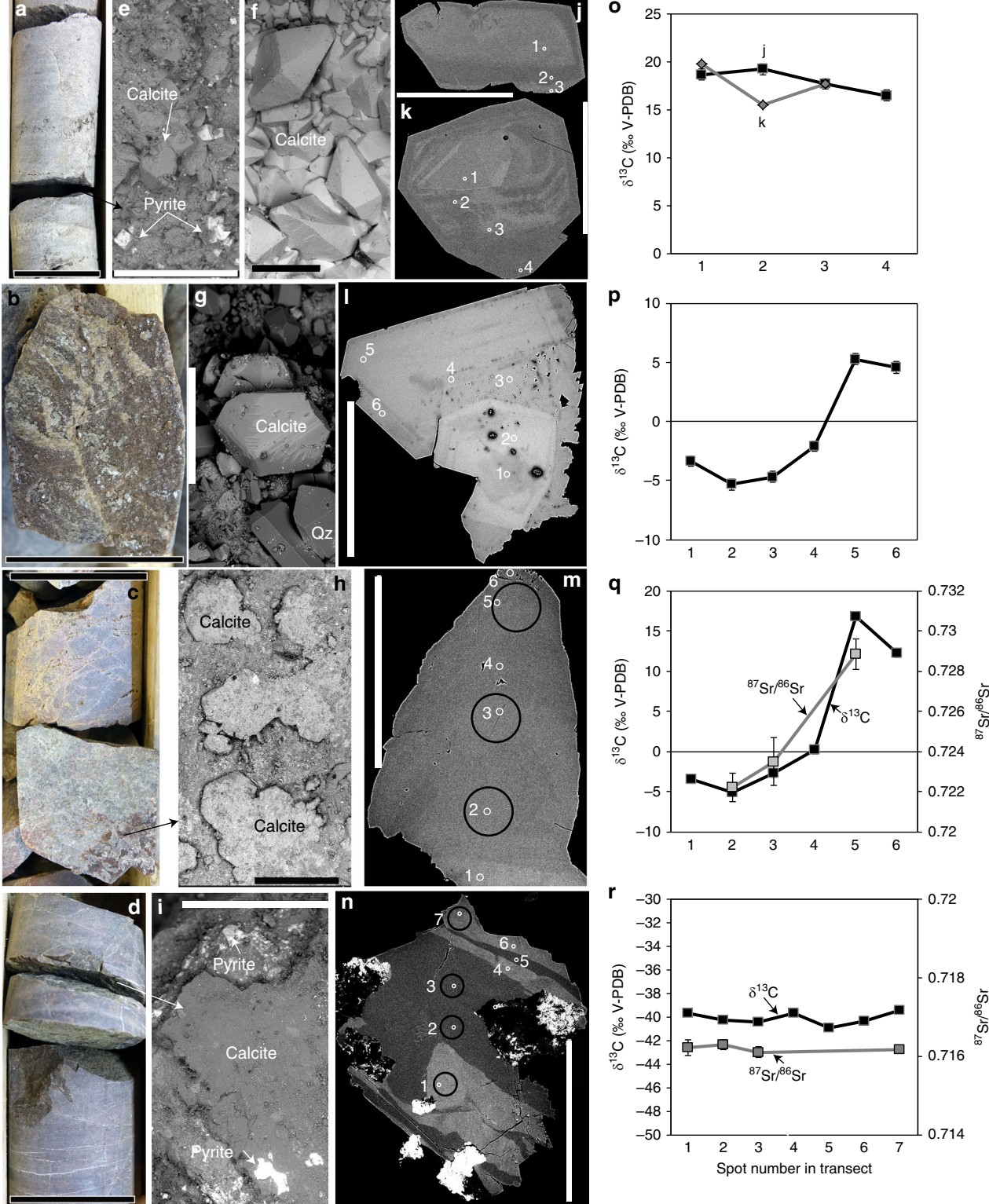

at the sediment-granite contact was $n$-$C_{17}$ to $n$-$C_{36}$, with a similar unimodal distribution as in the limestone, but with a peak maximum at $C_{25}$. Two of the granite samples (VM1:442 and CC1:539) had the same hydrocarbon distribution as in the rock contact, although the general abundance was lower and with distinct peaks of $n$-$C_{18}$, $n$-$C_{20}$ and $n$-$C_{22}$ and $n$-$C_{28}$. The deepest sample, CC1:608, showed hydrocarbons ranging from $n$-$C_{18}$ to $n$-$C_{32}$ with a maximum at $C_{20}$ and low abundances of $n$-$C_{26}$ to

$n$-$C_{32}$. Pristane (Pr) and Phytane (Ph) were present in the limestone and the two granite CC1 samples (Supplementary Data 7). The Pr/Ph, Pr/$n$-$C_{17}$ and Ph/$n$-$C_{18}$ ratios were relatively low (<1). In CC1:539, small amounts of $C_{27}$ to $C_{35}$ hopanoids, partly S and R isomers were detected (Fig. 6c). Fatty alcohols $n$-$C_{14}$-OH; $n$-$C_{16}$-OH, and $n$-$C_{18}$-OH were detected in all samples, in varying intensities (Fig. 6b). Mono ether lipid (1-o-n-hexadecylglycerol) was detected in VM1:251. The samples contained fatty acids (FA)

**Fig. 2** Appearance and paragenesis of calcite and transects of $\delta^{13}$C and $^{87}$Sr/$^{86}$Sr values. **a–d** Drill core photographs of fractured core sections with notable gas observations (according to drilling logs), SEM images of secondary minerals on the fracture surfaces (back-scatter electron [BSE] mode), (**e–i**), polished crystal cross-sections with transects of SIMS analyses indicated (**j–n**) and $\delta^{13}$C (**o–r**) and $^{87}$Sr/$^{86}$Sr values (**q, r**) corresponding to the spot locations in **j–n**. Details: **a** Sample VM2:170 m, limestone with open fractures. **e** Scalenohedral calcite crystals intergrown with cubic pyrite VM2:170 m. **f** Scalenohedral calcite crystals from a similar limestone fracture at a slightly greater depth, VM2:212 m. **j, k** polished crystals from **e** and **f**, respectively (corresponding $\delta^{13}$C values in **o**, all $^{13}$C-enriched). **b** Sample CC1:537, with euhedral calcite on a fracture in porphyritic crystalline rock coated by euhedral quartz (**c**). The calcite shows growth zonation (**l**) and is $^{13}$C-enriched in the outer parts (**p**). **c** Sample VM1:442 m, open fracture in heavily fractured crystalline rock section coated by aggregates of euhedral calcite (**h**). The calcite shows growth zonation (**m**) and is heavily enriched in $^{13}$C and $^{87}$Sr in the rim compared to the older growth zones (**q**). **d** Sample VM1:255 m, fractured section of crystalline rock with anhedral calcite coating with pyrite (**i**). The calcite shows growth zonation (**n**) but relatively homogeneous $^{13}$C-depleted $\delta^{13}$C values and $^{87}$Sr/$^{86}$Sr (**r**). Errors (2σ) are within the size of the symbols if not visible. Length of scale bars: (**a–d**) 4 cm, (**e, k**) 400 μm, (**f, h, i, j, n**) 500 μm, (**g, l, m**) 300 μm

ranging from $n$-C$_{12}$ to $n$-C$_{24}$, with a clear even over odd predominance and high abundances of $n$-C$_{16}$ and $n$-C$_{18}$ FA (Fig. 6b). Odd and branched FA$n$-C$_{15}$, $ai$-C$_{15}$, 12Me-C$_{16}$, $ai$-C$_{17}$, and $n$-C$_{17}$ and 12OH-C$_{18}$, as well as mono-unsaturated fatty acids C$_{18:1}$ and $n$-C$_{16:1}$ were also detected.

Time of flight secondary ion mass spectrometry (ToF-SIMS) performed to putative organic material in granite sample CC1:537 showed a negative spectra with peaks at m/z 255.2, 283.2, and 281.2 assigned to fatty acids C$_{16:0}$, C$_{18:0}$, and C$_{18:1}$[31], consistent with the GC-MS data. The positive spectra showed peaks that can be assigned to polycyclic aromatic hydrocarbon (PAHs, Supplementary Figs. 3 and 4).

**Composition of modern gas.** The gas encountered during borehole drilling in the sedimentary rock (VM2, gas and water samples) and in a borehole section covering sedimentary rock down below the granite contact (VM5, gas samples), is dominated by methane (mainly >90%, whereas CO$_2$ is at 3–14% in the gas samples, with even higher CO$_2$ in the water samples). The ratio of methane to higher hydrocarbons, C$_1$/(C$_2$ + C$_3$), is 125–200 (Supplementary Data 8) and there are notable relative concentration patterns: C$_2$ > C$_3$, iso-C$_4$ > $n$-C$_4$, $i$-C$_5$ > $n$-C$_5$, and $neo$-C$_5$ > $i$-C$_5$. The methane has $\delta^{13}$C values of −64 ± 2‰ which are lighter than the $\delta^{13}$C$_{C2}$ (−28 ± 2‰) and $\delta^{13}$C$_{C3}$ values (−7‰). The $\delta^2$H$_{CH4}$ value in the single sample analyzed for this ratio is −240‰SMOW for the VM5 borehole gas and $\delta^{13}$C$_{CO2}$ values are c. +5–8‰.

## Discussion

The highly variable $\delta^{13}$C$_{calcite}$ values between different fractures and within single crystals point to spatiotemporal variation of the processes that lead to calcite precipitation. We focus our discussion on processes producing the youngest calcites, which feature large $\delta^{13}$C$_{calcite}$ excursions. The older type predates the impact and dates back to 600–400 Ma (Supplementary Fig. 2) and shows no $\delta^{13}$C signatures diagnostic for methane cycling. In order to link the mineral data to the present gas in the system, the discussion starts with interpretations of the gas compositions that exist from the new boreholes and from previous prospecting.

Interpretation of the origin of hydrocarbon gases is typically based on diagnostic geochemical signatures, normally by using discrimination diagrams (Fig. 7, based on a global compilation[32]) and a holistic approach including the geological context. The most widely used discrimination diagram is the ratio of methane to higher hydrocarbons, C$_1$/(C$_2$ + C$_3$), versus $\delta^{13}$C$_{CH4}$ (Fig. 7a). This differentiates microbial gas which usually has high C$_1$/(C$_2$ + C$_3$) (>1000)[26] from the typically lower ratios of thermogenic methane (<50)[33,34]. However, abiotic gas may also show high C$_1$/(C$_2$ + C$_3$)[35,36] and cannot be excluded based on this ratio. For $\delta^{13}$C$_{CH4}$, there is typically a difference between methane sources, ranging from the substantially $^{13}$C-depleted microbial, through moderately $^{13}$C-depleted thermogenic to isotopically heavier abiotic methane[26]. Microbial methanogenesis can be divided into

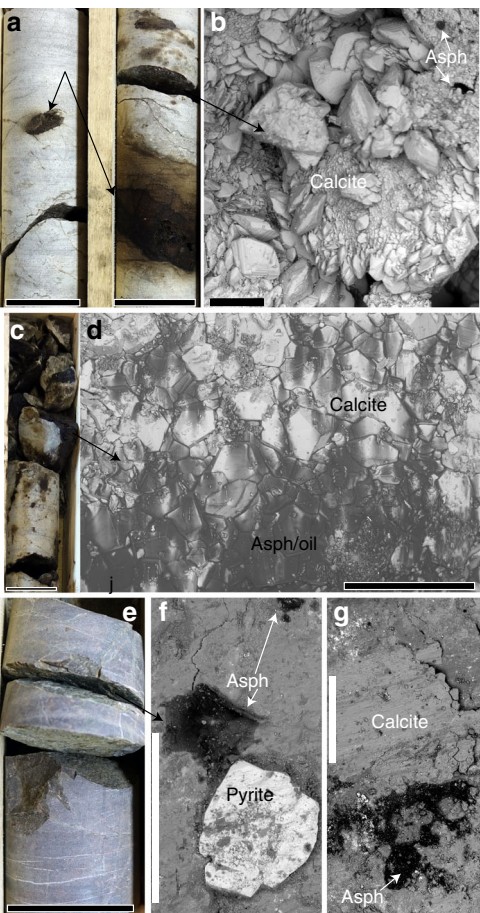

**Fig. 3** Appearance and paragenesis of solid bitumen and seep oil. (**a–d**) from sedimentary rock fractures (**e–g**) from crystalline rock fractures. Drill core photographs are shown in (**a, c, e**) and SEM images of secondary minerals on the fracture surfaces (BSE mode) are shown in (**b, d, f, g**). **a** White limestone (VM2:328 m) with abundant solid bitumen (asphaltite = "Asph") and oil seep. **b** Abundant scalenohedral calcite ($^{13}$C-rich) crystals on the fracture in (**a**), and small patches of bitumen ("Asph"). **c** Fractured section of limestone (01–10C:326 m) with abundant occurrences of solid bitumen and oil seep. **d** Euhedral calcite ($^{13}$C-rich) crystals on the fracture in (**a**) with abundant bitumen and seep oil smeared on the surface. **e** Drill core sample VM1:255 m (fractured crystalline rock). **f** Euhedral pyrite and bitumen on the fracture surface (**e**). **g** Bitumen ("Asph") and calcite (slickensided and partly euhedral crystals) in an adjacent fracture, VM1:251 m. Length of scale bars: (**a, c, e**) 4 cm, (**b, f, g**) 500 μm, (**d**) 1 mm

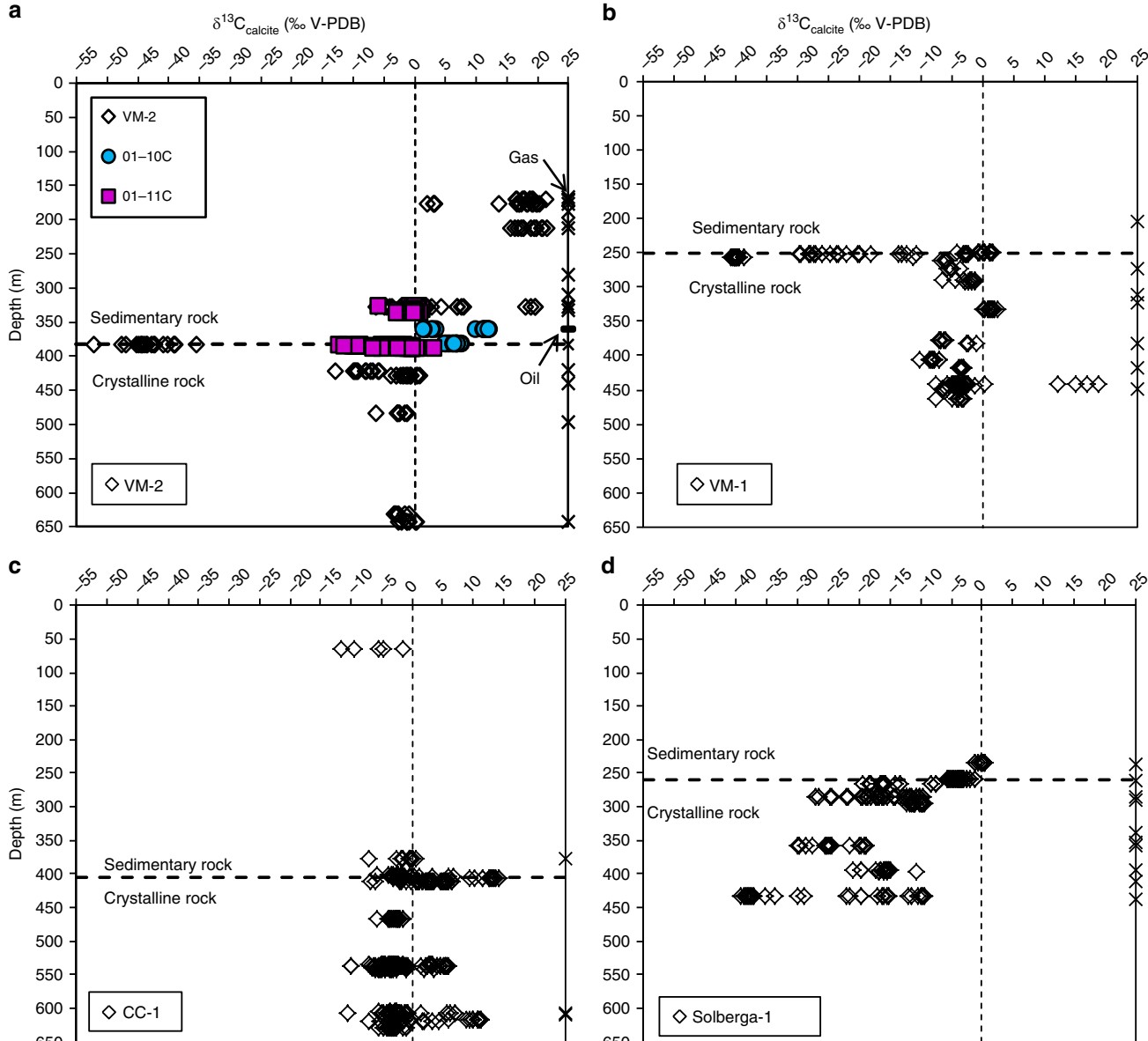

**Fig. 4** $\delta^{13}C_{calcite}$ vs depth. **a** Closely spaced boreholes VM2, 01–10C and 01–11C (samples from the latter two have been depth-normalized to the sedimentary-crystalline rock interface in VM-2 in order to plot them in the same graph), (**b**) VM1, (**c**) CC1, and (**d**) Solberga-1, with Solberga quarry samples collected at close to ground surface, ~400 m south of Solberga-1 drill site. Each data point represents one SIMS analysis. Fractures with gas occurrences observed during drilling are marked with "x" (including a seep oil observation from borehole 01–10C/VM3, marked "−") on the right-hand side of the graphs and the depth of the sedimentary-crystalline bedrock interface is also indicated. Errors (2σ) are within the size of the symbols. All plots have the same range on the axes for comparison

the carbonate reduction pathway and acetate (methyle-type) fermentation[37], of which the former involves a larger kinetic carbon isotope effect[26]. At Siljan, a dominantly microbial gas fraction is suggested by the light $\delta^{13}C$ values of the methane (−64 ± 2‰, Fig. 7a). However, the $C_1/(C_2 + C_3)$ values are slightly lower than expected from a pure microbial gas and therefore point to a mixed origin, as indicated by the position at the border between the microbial carbonate reduction and early mature thermogenic fields (Fig. 7a). Regarding the higher hydrocarbons, it has been demonstrated that microbial ethano- and propanogenesis occur in deeply buried marine sediments[38], and the former also near gas wells in western Canada[39]. Presence of ethane and propane is thus not a definite marker for thermogenic gas. However, in a microbial gas, the presence of $C_{4+}$ gases (detectable $n$-C$_4$, $i$-C$_4$, $i$-C$_5$, Supplementary Data 8) is not

expected[33,34], and the $\delta^{13}C_{C2}$ values are typically not as heavy as those measured (−28 ± 2‰)[40], in particular in comparison to the light $\delta^{13}C_{CH4}$ values (−64 ± 2‰), which indicate thermogenic contribution. These relatively heavy $\delta^{13}C_{C2}$ values and even heavier $\delta^{13}C_{C3}$ values speak against a significant contribution from abiotic gas, which generally features decreasing $\delta^{13}C$ values with higher carbon number of the homologues[41]. At other igneous rock sites in South Africa, Canada, and Scandinavia, abiotic methane shows lighter $\delta^{13}C$ values (−50‰[10,41]) than the typically assigned values of abiotic methane (i.e. >−20‰[42]), although not as low as the methane at Siljan.

The isotopic hydrogen signature ($\delta^2H_{CH4}$) of −240‰ SMOW for the VM5 borehole gas is, when plotted against $\delta^{13}C_{CH4}$ (Fig. 7b) also in the microbial carbonate reduction field[32], however, close to early mature thermogenic gas and

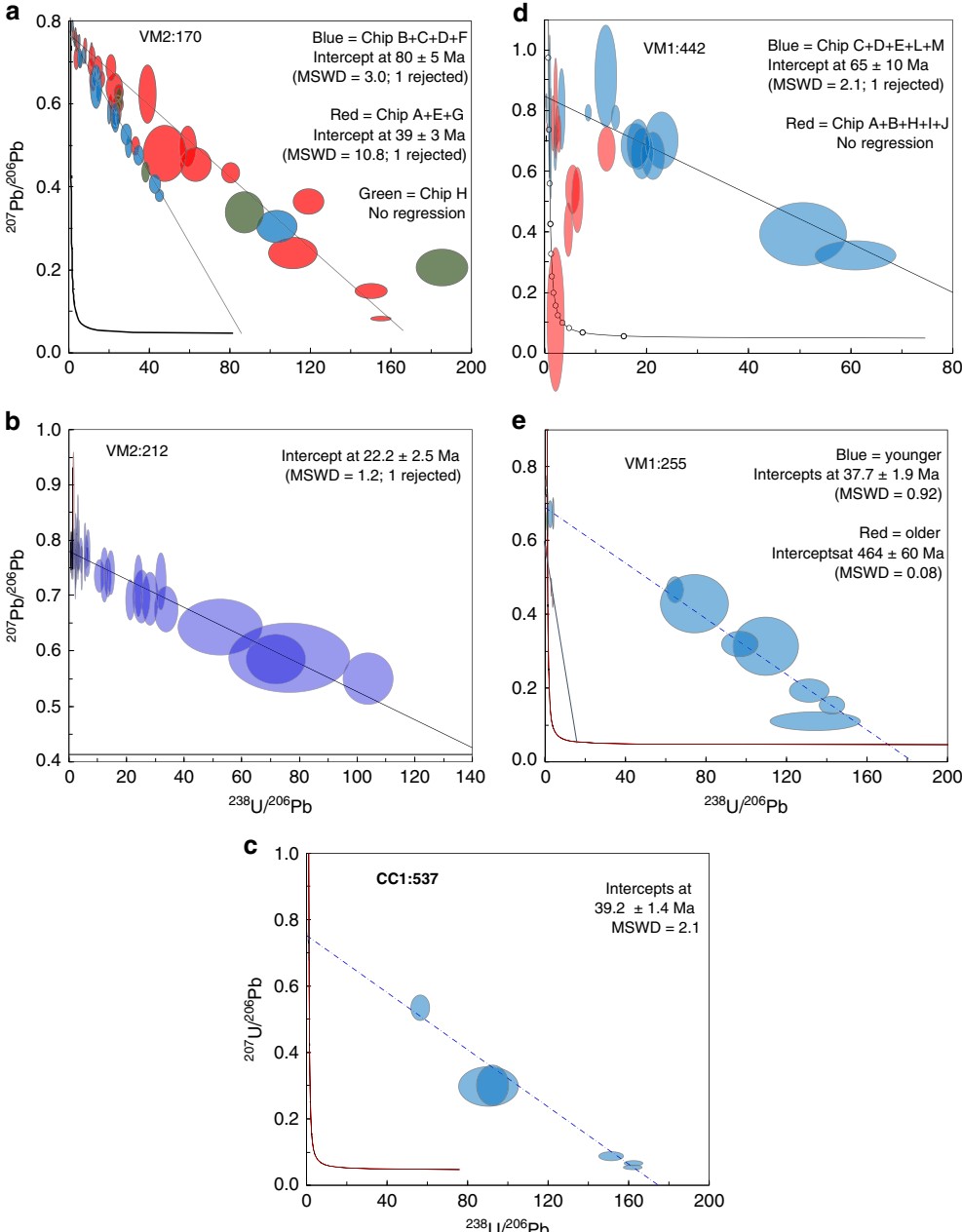

**Fig. 5** U-Pb carbonate dating. **a** VM2:170, two generations of [13]C-rich calcite in limestone fracture, yielding ages 80 ± 5 Ma and 39 ± 3 Ma. **b** VM2:212 [13]C-rich calcite in limestone fracture, yielding 22.2 ± 2.5 Ma. **c** CC1:537, [13]C-rich calcite, in granite fracture, yielding 39.2 ± 1.4 Ma. **d** VM2:442, [13]C-rich calcite in granite fracture, yielding 65 ± 10 Ma. **e** VM1:255, [13]C-depleted calcite in granite fracture (targeting the growth zones with dark BSE-intensity in Fig. 2n), yielding 37.7 ± 1.9 Ma, with an older potential event 464 ± 60 Ma (highly uncertain: only three data points). Errors represented by the ellipses are 2σ

fermentation type microbial gas. Additionally, the Siljan gas samples are overlapping with $\delta^2H_{CH4}$ ranges of abiotic sources at other sites[10], meaning that abiotic contribution cannot be ruled out based on the $\delta^2H_{CH4}$ composition. Overall, the position of the gas samples at borders or within multiple empirically defined zones on the discrimination plots (Fig. 7a, b) shows that these plots alone are not diagnostic for any single process and/or gas origin.

The heavy $\delta^{13}C_{CO2}$ (c. + 5–8‰, Fig. 7c, Supplementary Data 8) of the Siljan samples is typical for microbial methanogenesis through carbonate reduction[32] and thus another feature supporting a dominantly microbial gas origin. These $\delta^{13}C_{CO2}$ values are characteristic for secondary microbial methane[43] formed following microbial utilization of primary thermogenic

hydrocarbons (e.g. petroleum, seep oils and lighter hydrocarbons), which is supported by other biodegradation signatures. These signatures include high $C_2$ to $C_3$ ratios owing to that ethane is relatively resistant to biodegradation compared to the $C_{3+}$ homologues[44]. Biodegradation also discriminates against $^{13}C_{C3}$, leading to isotopically heavy residual propane[44]. In the Siljan gas, the anomalously heavy $\delta^{13}C_{C3}$ values compared to the $\delta^{13}C_{C2}$ values, the high $C_2$ to $C_3$ ratios that are far from the normal range for thermogenic gases, as well as other signatures presented in Supplementary Note 1, thus point to biodegradation, but to an unknown degree. The removal of the higher hydrocarbons during biodegradation increases the $C_1/(C_2\text{-}C_3)$, which complicates the estimation of the mixing proportions between microbial and thermogenic gas.

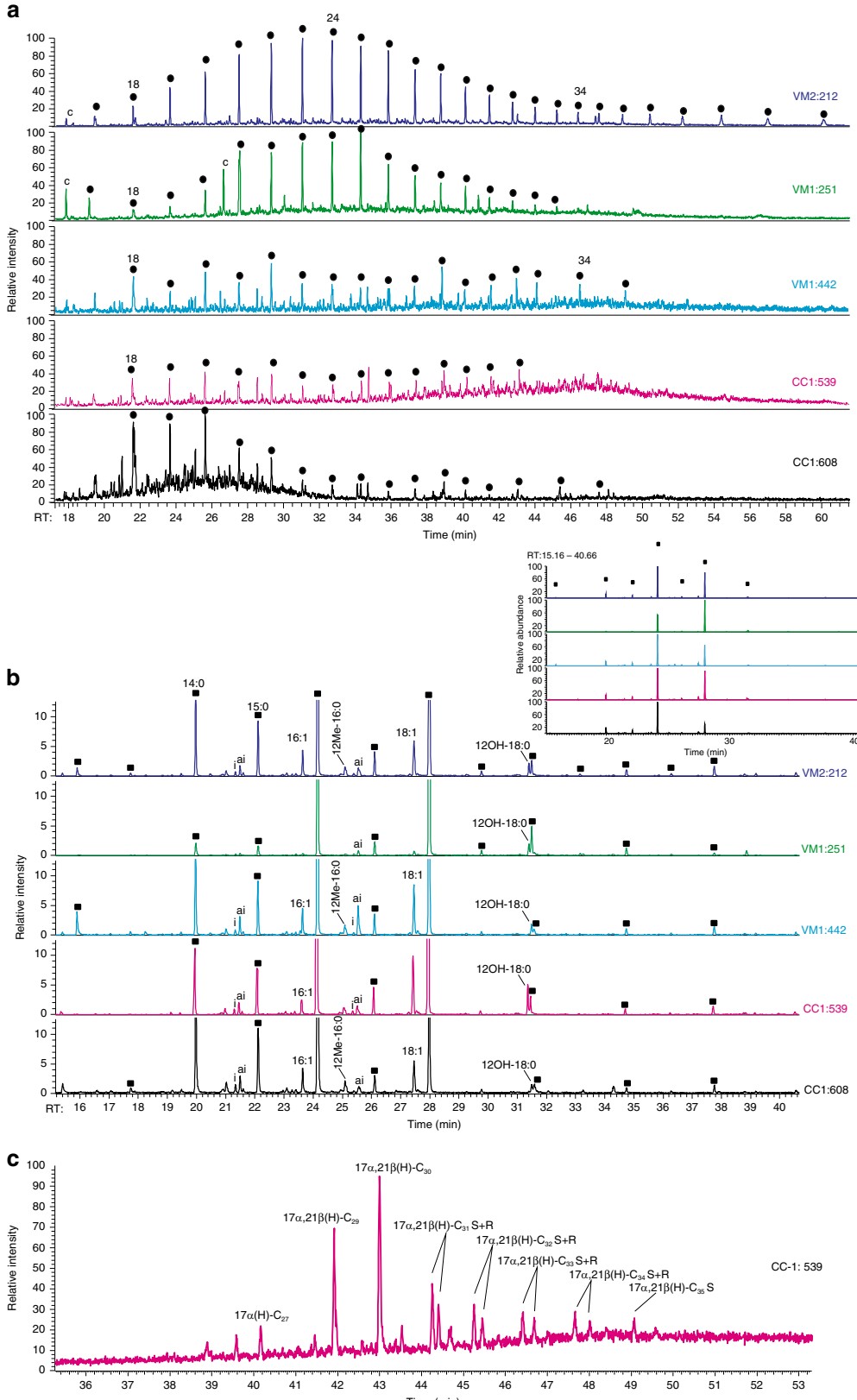

**Fig. 6** Partial mass chromatograms. Samples are from limestone fracture (VM2:212), sediment-granite contact (VM1:251) and deep granite fractures (VM1:442, CC1:539, CC1:608), showing (**a**) the *n*-alkane distribution pattern (straight chain hydrocarbons, m/z 85, ●), (**b**) the fatty acid distribution (m/z 69, 74; ■) of all investigated samples and (**c**) weak hopanoid signals (m/z 191) from the calcite in the granite fracture sample CC1:539

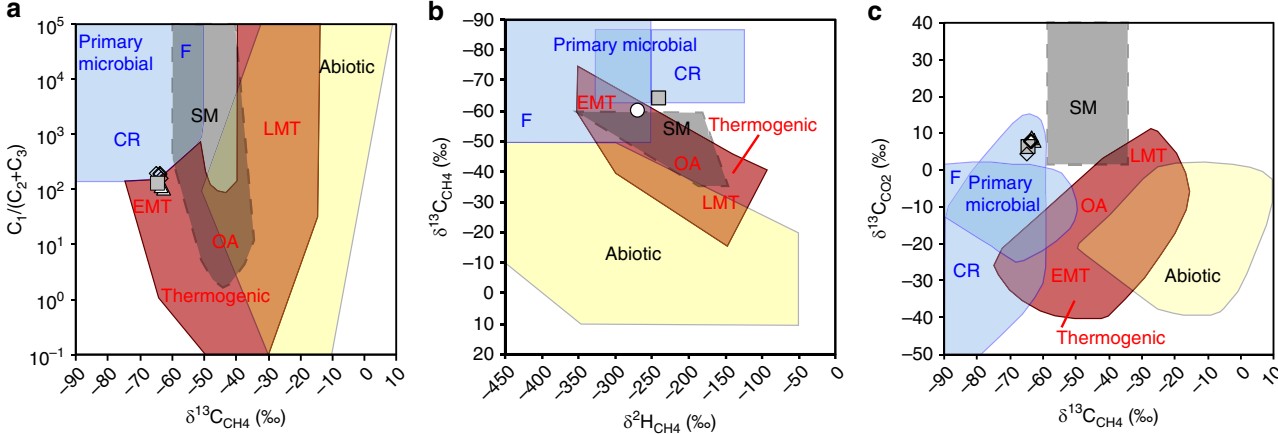

**Fig. 7** Gas composition discrimination diagrams. (**a**) $\delta^{13}C_{CH4}$ versus $C_1/(C2 + C_3)$. (**b**) $\delta^2H_{CH4}$ versus $\delta^{13}C_{CH4}$. (**c**) $\delta^{13}C_{CH4}$ versus $\delta^{13}C_{CO2}$ (adapted from[32]). Position of the gases from boreholes VM2 (diamonds = gas samples, triangle = water samples), VM5 (square) and drinking water well (circle) are shown. Genetic gas field abbreviations denote: CR $CO_2$ reduction, F methyl-type fermentation, SM secondary microbial, EMT early mature thermogenic, OA oil-associated thermogenic, LMT late mature thermogenic gas. Gas data were extracted from the database of AB Igrene (Supplementary Data 8), and from[45] (drinking well at Gulleråsen close to Solberga, $\delta^{13}C_{CH4}$: −60.3‰, $\delta^2H_{CH4}$: −269‰). The sampling site of the gas data in borehole VM2 corresponds to the uppermost $^{13}C$-rich calcites in this borehole, whereas the other gas-sampled borehole (VM5) is just adjacent to other boreholes sampled for calcite (VM- and 01-boreholes). Errors (2σ) are within the size of the symbols

All the gas data together thus point to a gas that is to a large extent microbial and to a significant extent thermogenic. Although abiotic gas contribution cannot be directly identified in the investigated (dominantly sedimentary) aquifer, it cannot be ruled out, at least not in the deep granite fracture system, because none of the borehole samples isolates gas from the crystalline aquifer alone. Theoretically, abiotic gas contribution from the granite fractures may thus be masked by gases from the sedimentary rock fractures. Variably depleted $\delta^{13}C_{CH4}$ signatures from previous investigations[16,45], summarized in Supplementary Note 2, are in accordance with the mixed gas of interpreted dominantly microbial and thermogenic origin detected in the present study (Fig. 7). Hence, although the results presented here and previously are not generally supportive of abiotic gas, such gas cannot be fully ruled out, at least not in the deeper, granitic system.

For the mineralogical record, the significantly $^{13}C$-enriched calcite observed in the fractures in limestone ($\delta^{13}C_{calcite}$ values as heavy as +21.5‰, Fig. 4b) and granite (up to +18.9) is evidence for formation following microbial methanogenesis in situ, owing to the discrimination that occurs against $^{13}C$ during methanogenesis that leaves $^{13}C$-rich residual carbon behind[25,27,46]. However, the presence of $^{13}C$-rich calcite cannot completely rule out minor abiotic gas fractions. In addition, the FA $n$-$C_{12}$ to $n$-$C_{18}$, particularly the odd chain and branched $iC_{15}$, $aiC_{15}$, $n$-$C_{15}$, $12Me$-$C_{16}$, $aiC_{17}$, and $12OH$-$C_{18}$ as well as the $n$-alcohols and the 1-o-n-hexadecylglycerol preserved within $^{13}C$-rich, methanogenesis-related, calcite coatings are support for in situ microbial activity. These preserved FA can be tied to fermentation[47], sulfate reduction by bacteria[48] and other microbial processes (Supplementary Note 3), but not specifically to methanogens (*archaea*), which do not produce phospholipid fatty acids.

The U-Pb ages suggest that methanogenesis in the sedimentary and granite aquifers at Siljan led to precipitation of $^{13}C$-rich calcite on several occasions, from 80 ± 5 to 22.2 ± 2.5 Ma (Fig. 5). The distribution of this calcite marks microbial methanogenesis in the upper 214 m of the fractured crystalline rock in the crater structure (to depths of 620 m) and in the overlying sedimentary rock fractures over a depth span of more than 200 m. The substantial $^{13}C_{calcite}$ and $^{13}C_{CO2}$ enrichments occurring in the limestone aquifer are noteworthy, because dilution by the C isotope

signature of dissolved inorganic carbon (DIC) derived from limestone ($\delta^{13}C$: 0 to +2‰[49]) would be expected. To explain this feature, we propose local influence from kinetic microbial processes on the $\delta^{13}C_{DIC}$ signature in the Siljan aquifer, in common with observations from other deep energy-poor fracture system[28,50] (Supplementary Note 4). This phenomenon should be particularly important in pore space infiltrated by gases, bitumen or seep oils, as shown by spatial relation of these features to significantly $^{13}C$-rich calcite (Fig. 3). Preserved hydrocarbon $n$-alkane pattern of calcite in bitumen-bearing fractures of the sedimentary rock and at the sediment-granite interface (VM2:212; VM1:251, Fig. 6) is indication for thermal- and biodegradation. It has previously been reported that biomarkers in bitumen in sedimentary rock fractures link its origin to shales and that mobilization and degradation of hydrocarbons have occurred on several events in the fracture systems[22] (additional biomarker support in Supplementary Note 1).

Methanogenesis is commonly associated with sulfate-poor biodegraded petroleum reservoirs[51] and initial steps of anaerobic utilization of organic matter (fermentation) involve hydrolysis followed by bacterial acetogenesis that converts volatile fatty acids into acetic acid, $H_2$ and $CO_2$[52]. Alternatively, $H_2$ is produced by aromatization of compounds present in the seep oil[51]. Methanogenesis through $CO_2$ reduction, with $H_2$ as electron donor, has been proposed to be the dominant terminal process in petroleum biodegradation in the subsurface[47], and this appears also to be the case at Siljan based on the widespread and pronounced heavy $\delta^{13}C_{calcite}$ and $\delta^{13}C_{CO2}$ values (Figs. 4 and 7c). In sulfate-rich reservoirs, microbial sulfate reduction (MSR) can be involved in degradation of hydrocarbons. In the Siljan fractures, pyrite occasionally occurs together with $^{13}C$-enriched calcite. Pyrite formed by MSR is typically strongly depleted in $^{34}S$[53]. The very low minimum $\delta^{34}S_{pyrite}$ values (−40‰V-CDT, Supplementary Fig. 1) is thus proposed to reflect MSR. However, groundwater in granite fractures of adjacent boreholes show very low sulfate concentrations, 4.3–6.6 mg L$^{-1}$[45], suggesting a generally low potential for MSR in that aquifer. Although anaerobic oxidation of organic matter by MSR can produce $CO_2$ that can be utilized by methanogens[43], it did probably not result in large quantities of methane because sulfate reducers outcompete methanogens for $H_2$ and other substrates when sulfate concentrations are

elevated[54]. Instead, fermentation likely dominated initial degradation steps of organics in the system providing $H_2$ for the methanogens to perform reduction of $CO_2$. Furthermore, the low salinity in the deep granite aquifer[45] is favorable for microbial methanogenesis[55]. Secondary methane formation following microbial utilization of primary thermogenic hydrocarbons typically involves large $^{13}C$-enrichment in carbonate[32], as manifested by heavy $\delta^{13}C_{calcite}$ (Fig. 4) and $\delta^{13}C_{CO2}$ (Fig. 7c).

Overall, the geological setting with relatively low temperatures and shallow reservoir with abundant seep oil/bitumen are, together with gas signatures and $^{13}C$-enriched calcite, in favor of formation of secondary microbial methane produced following biodegradation of thermogenic hydrocarbons (gas, seep oil and bitumen). The primary thermogenic gas remains as a minor biodegraded mixing fraction, in common with secondary methane reservoirs elsewhere[32,43]. In the deeper granite system, contribution from abiotic gas sources may also have been involved.

During AOM, a phenomenon where methanotrophs can act in syntrophic relationship with MSR, carbonate may precipitate and inherit the significantly $^{13}C$-depleted signatures of the methane[56]. The light $\delta^{13}C_{calcite}$ values detected at the sediment-granite contact at Siljan (−52.3‰, Fig. 4) are thus proposed to reflect AOM (Supplementary Note 5 describes more moderately $^{13}C$-depleted calcite). The U-Pb age of this calcite shows that AOM dates back at least 39 ± 3 Myr. The $\delta^{13}C_{calcite}$ values point to utilization of methane of dominantly microbial origin[10,26], because thermogenic and abiotic methane are usually heavier[57]. The $\delta^{13}C$ signature of carbonate originating from oxidized methane is typically diluted by other relatively $^{13}C$-rich dissolved carbon species prior to incorporation in calcite[29]. When taking such dilution into account, it is likely that the source methane was isotopically light, in line with the $\delta^{13}C_{CH4}$ composition (−64 ± 2‰) of dominantly microbial origin (with a minor thermogenic and possibly a minor abiotic component) in boreholes VM2/5. Furthermore, in the sample with the most $^{13}C$-depleted (AOM-related) calcite, co-genetic pyrite has low minimum $\delta^{34}S_{pyrite}$ values (−18.7‰, Supplementary Fig. 1) reflecting large $^{32}S$ enrichment characteristic for MSR-related sulfide[53]. This finding, together with MSR-related[58,59] branched fatty acids ($ai$-$C_{15:0}$, 12Me-$C_{16:0}$, $ai$-$C_{17:0}$, 12OH-$C_{18:0}$, Fig. 6b) mark coupled AOM-MSR at the sedimentary-granite rock contact. Another noteworthy feature is the overall large $\delta^{34}S_{pyrite}$ spans that mark MSR-related reservoir effects throughout the fracture system (Supplementary Fig. 5 and Note 5).

Several precipitation events are recorded by the intra-crystal variability of the C and Sr isotopes, and the U-Pb age groups. Our interpretation is that these precipitation events were caused by fracture reactivations, as presented in detail in Supplementary Note 6. In summary, there are tectonic events in the far-field and uplift events that temporally coincide with the ages of the methane related calcite at Siljan. Methane cycling can thus be related to these fracture reactivation events that are more than 300 million years younger than the impact.

A conceptual model for methane accumulation in the Siljan impact structure, as outlined in Fig. 8, has its basis in the isotopic inventory of secondary fracture minerals and gases. The microbial methanogenic processes date back at least to the Late Cretaceous (Fig. 5), although there are fractures in the granite that formed significantly earlier (Supplementary Fig. 2). The gas compositions corroborate that the gas in the sedimentary reservoir is microbial with contribution from a biodegraded thermogenic end-member linked to thermal maturity of black shales in the sedimentary pile, and perhaps a minor abiotic gas fraction. In the upper part of the sedimentary successions there is local seepage to the surface, as shown by methane in drinking water wells[45]. The isotopic composition of methane in such a well[45] fits with both microbial and early mature thermogenic origin (Fig. 7b). Bitumen can be mobilized when thermally affected[60]. Bitumen and seep oil migration from the organic-rich shales into other sedimentary rock units and into the fractured granitic basement have thus likely been initiated when the sediments were thermally matured, either as a result of the heat from the impact[23] or due to subsidence related to Caledonian foreland basin crustal depression[61]. The bitumen and seep oil occurrences (along with migrated thermogenic gas) provided energy for the indigenous microbial communities in the deep subsurface, as shown by the spatial relation to $^{13}C$-rich calcite. Deep abiotic gas contribution to the methane accumulations in the granite fractures cannot be ruled out. However, the apparently higher (but not yet quantified) abundance of gas beneath the sedimentary rock in the crater rim (Fig. 4) than in the central dome[16], the input of shale-derived hydrocarbons to the granite fractures, and the similar $^{13}C$-enrichment of calcite in granite and sedimentary fractures point to similar formation and accumulation of methane in the granite fracture network as in the sedimentary rock. The dominantly Eocene–Miocene ages of the $^{13}C$-rich calcite indicate that the major microbial utilization of the hydrocarbons in the deep fractures occurred when temperatures were more favorable (<50 °C) for microbial activity, in line with the uplift and subsidence history of the south-central Fennoscandian shield[62]. The Eocene–Miocene microbial activity is proposed to be linked to regional re-opening of bitumen-bearing fracture sets. This enabled circulation of groundwater along flow paths with substrates accessible to the microbes in the form of bitumen/oil coatings, as well as facilitated circulation of biodegradable thermogenic gas in the deep reservoir. The spatial relation of $^{13}C$-enriched calcite and biodegraded bitumen/seep oil suggests secondary methane formation following anaerobic degradation of organic matter. This fermentation process produces $H_2$ for utilization by methanogens through reduction of $CO_2$ formed during biodegradation or occurring in the aquifer. The kinetic microbial processes producing methane resulted in large isotopic fractionations, as observed in the gases and secondary carbonates. Taken together, there are numerous lines of evidence in favor of long-term microbial methane formation in the Siljan crater, likely fueled by thermogenic gas, seep oil and bitumen mobilized from shales in the sedimentary successions and transported through fracture conduits to the deeper granite aquifer. The sedimentary successions, in turn, acted as cap rocks for the gas in the granite fractures.

Input of hydrocarbons to the deep microbial communities has potential to result in accumulation of methane in basement fracture networks beneath sedimentary cap rocks. A relationship between $^{13}C$-rich calcite and bitumen like at Siljan occurs in deep crystalline rock fractures at Forsmark, Sweden[25] and solid and gaseous hydrocarbon occurrences of sedimentary origin occur in fractured crystalline basement rocks on the British Isles[63], Australia[64], and the United States[65]. Whereas microbial generation of economic accumulations of methane within organic-rich shale are known from several locations[66], the extent of gas accumulations in the upper crystalline continental crust buried beneath sedimentary successions and in fractured impact structures are less explored. The upper crystalline continental crust environment makes up one of the largest, but yet least surveyed, deep biosphere habitats on Earth. The extent, continuity and physicochemical prerequisites for gas accumulation here require more attention in order to assess the significance of this underexplored greenhouse gas source on a global scale.

In the Siljan impact structure, a relation between methane cycling and deep subsurface life is evident. The physical influence of the actual impact and the long-term effects are manifested by,

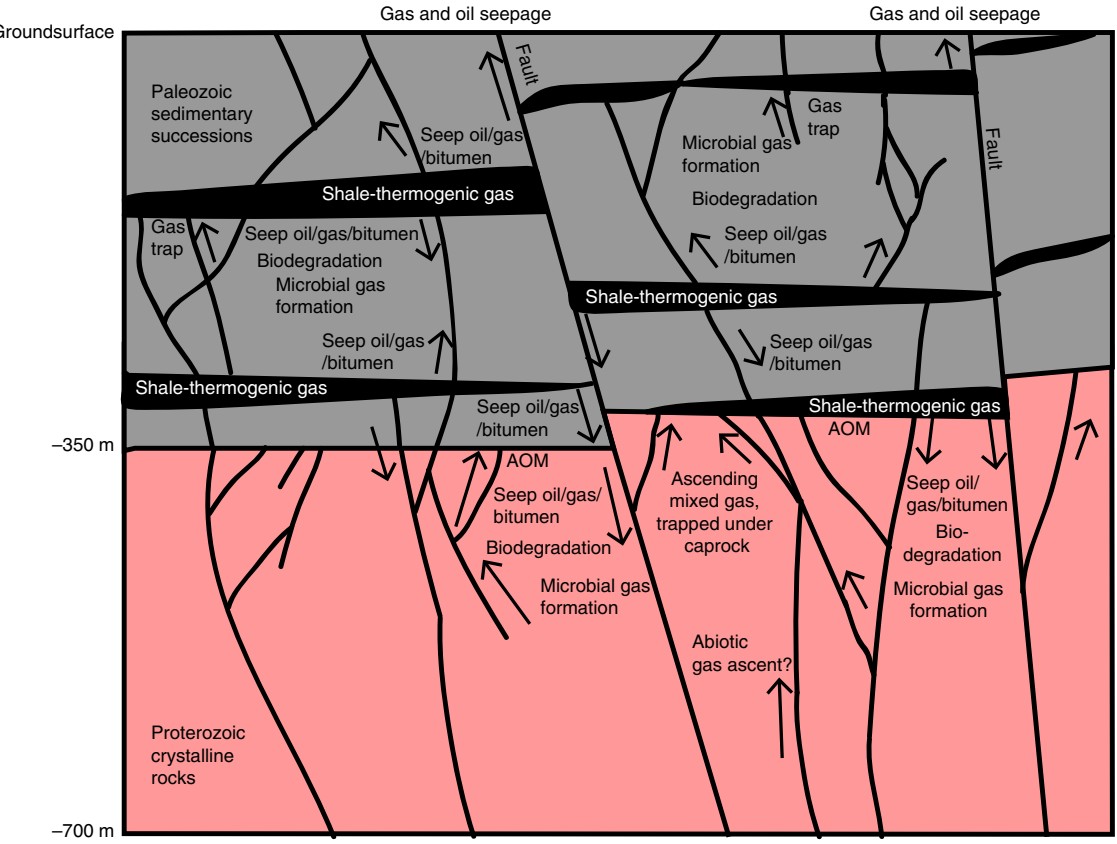

**Fig. 8** Conceptual model of the gas accumulation in the Siljan ring impact structure. Thermogenic gas formed in the (to varying degree) mature Silurian and Ordovician black shales in the sedimentary strata. This gas and related seep oil and bitumen dispersed in the adjacent sedimentary successions with local surficial seepage. Downward migration of these hydrocarbons has occurred into the granitic basement during fracture reactivation events. Biodegradation of the hydrocarbons has occurred in the fracture system and secondary methane has formed in situ. The mixed gas, of microbial (dominantly) and biodegraded thermogenic type, which also may have an abiotic end-member, has accumulated at the sedimentary-granite contact where anaerobic oxidation of methane has occurred

first, abundant fracturing compared to surrounding rocks[15] which is particularly important in igneous lithologies where colonization is restricted by the pore space[2]; second, dislocation of organic-rich sedimentary rocks that provide pathways for surficial organics to the deep endolithic communities; and third, development of a cap rock enclosing methane at depth. These effects collectively enable microbial colonization of the crater hundreds of millions of years after the impact. Our findings of widespread long-lasting deep microbial methane-forming communities in the Siljan crater support the hypothesis that impact craters are favorable for deep microbial colonization[2,3]. However, the link between microbial methanogenesis and organics from Paleozoic shale challenges the use of the Siljan crater as an analog for instantaneous extra-terrestrial impact-related colonization. In an astrobiological context, the Siljan crater findings are nevertheless of large importance, as they display that multi-disciplinary micro-scale constraints for microbial activity (stable isotopes, geochronology, biomarkers) are needed to confirm that colonization and impact in ancient crater systems are coeval. This is particularly important because post-impact microbial colonization will likely occur in these favorable deep microbial settings and can thus easily be misinterpreted as impact related. Finally, the methods we have used here to provide the first evidence of long-term microbial methane formation and accumulation in a terrestrial impact crater would be optimal to apply to other impact-crater fracture systems, including methane emitting craters on Mars[67], in order to enhance the understanding of microbial activity and gas cycling in this underexplored environment.

## Methods

**Materials**. In this study, secondary calcite and pyrite coatings were collected from open fractures in cores from a total of seven boreholes at Mora ($n = 5$), Solberga and Stumsnäs (Fig. 1) drilled in 2011–2018. Samples are from 64 to 642 m vertical depth from the ground surface. The co-genetic calcite and cubic pyrite crystals occur on the fracture surfaces in the sedimentary rocks (Fig. 2a) and crystalline basement (Fig. 2b–d). Calcite mainly occurs as subhedral flat aggregates (Fig. 2i) to euhedral crystals of scalenohedral (Fig. 2e, f, m) or short c-axis type (Fig. 2g). Polished cross-sections reveal growth zonation in several calcite crystals (Fig. 2j–n). The paragenesis includes clay minerals, harmotome, apophyllite, sphalerite, galena and quartz but most of these minerals are related to the oldest growth phases of calcite. Seep oil and solid bitumen are present, particularly in the sediments (e.g. at 326 and 382 depth, where oil covers and is intermixed with calcite, Fig. 3a–d) but are also present in the crystalline basement fractures (Fig. 3e–g). Additional samples were taken from fractured Upper Ordovician Boda limestone in a quarry at Solberga. All boreholes are drilled through an upper layer of Paleozoic sedimentary rocks into the Proterozoic crystalline bedrock. The interface between the rock units is at between 250 to 406 m depth at Mora (borehole VM1: 250.9 m; VM2: 382 m; 01–10C: 346 m; 01–11C: 400.5 m and CC1: 406 m) and Solberga (Solberga-1: 259 m). At Stumsnäs, large scale impact-related faulting has caused a slab of Proterozoic granite to overthrust the sedimentary successions which thus are sandwiched in between blocks of granite at 196–286 m depth[69,70]. Samples were taken from fractures in the sedimentary rocks (limestone and shale) and from the upper 212 m of the underlying crystalline basement. Sampling focused, but was not limited, to borehole sections with elevated gas concentrations as observed during drilling. The mineralogy and appearance of the uncoated fracture coatings were examined under low-vacuum conditions in a Hitachi S-3400N Scanning Electron Microscope (SEM) equipped with an integrated energy dispersive spectroscopy (EDS) system. The coatings were then scraped off for various analyses (fluid inclusions, stable isotopes, radioisotopes and biomarkers).

Gas from isotube sampling and gases in waters from isojar sampling have been analyzed by AB Igrene for chemical composition from a few deep borehole sections (Supplementary Data 8).

**SIMS δ13C, δ18O, δ34S.** Calcite and pyrite crystals were mounted in epoxy, polished to expose cross-sections and examined with SEM to trace zonations and impurities prior to SIMS analysis. Intra-crystal SIMS analysis (10 μm lateral beam dimension, 1–2 μm depth dimension) of sulfur isotopes in pyrite and carbon and oxygen isotopes in calcite was performed on a Cameca IMS1280 ion microprobe at NordSIM, Swedish Museum of Natural History, Stockholm, following the analytical settings and tuning reported previously[28]. Sulfur was sputtered using a $^{133}Cs^+$ primary beam with 20 kV incident energy (10 kV primary, −10 kV secondary) and a primary beam current of ~1.5 nA. A normal incidence electron gun was used for charge compensation. Analyses were performed in automated sequences, with each analysis comprising a 70 s pre-sputter to remove the gold coating over a rastered $15 \times 15$ μm area, centering of the secondary beam in the field aperture to correct for small variations in surface relief and data acquisition in sixteen four second integration cycles. The magnetic field was locked at the beginning of the session using an NMR field sensor. Secondary ion signals for $^{32}S$ and $^{34}S$ were detected simultaneously using two Faraday detectors with a common mass resolution of 4860 (M/ΔM). Data were normalized for instrumental mass fractionation using matrix matched reference materials which were mounted together with the sample mounts and analyzed after every sixth sample analysis. Results are reported as per mil (‰) δ34S based on the Canon Diablo Troilite (V-CDT)-reference value. Analytical transects of up to seven spots were made from core to rim in the crystals. Up to seventeen crystals were analyzed from each fracture sample. In total, 443 analyses were made for δ34S ($^{34}S/^{32}S$) of pyrite from 115 crystals from nineteen fracture samples. The pyrite reference material S0302A with a conventionally determined value of 0.0 ± 0.2‰ (R. Stern, University of Alberta, pers. comm.) was used. Typical precision on a single δ34S value, after propagating the within run and external uncertainties from the reference material measurements was ±0.10‰.

For calcite, a total number of 984 δ13C (17 for δ18O) SIMS analyses were performed. Settings follow those described for S isotopes, with some differences: O was measured on two Faraday cages (FC) at mass resolution 2500, C used a FC/EM combination with mass resolution 2500 on the $^{12}C$ peak and 4000 on the $^{13}C$ peak to resolve it from $^{12}C^1H$. Calcite results are reported as per mil (‰) δ13C based on the Pee Dee Belemnite (V-PDB) reference value. Analyses were carried out running blocks of six unknowns bracketed by two standards. Analytical transects of up to nine spots were made from core to rim in the crystals. Up to fifteen crystals were analyzed from each fracture sample. Analyses were made for 331 crystals from 67 fracture samples (50 from granite and 17 from sedimentary rock). Isotope data from calcite were normalized using calcite reference material S0161 from a granulite facies marble in the Adirondack Mountains, kindly provided by R.A. Stern (Univ. of Alberta). The values used for IMF correction were determined by conventional stable isotope mass spectrometry at Stockholm University on ten separate pieces, yielding δ13C = −0.22 ± 0.11‰V-PDB (1 std. dev.) and δ18O = −5.62 ± 0.11‰ V-PDB (1 std. dev.). Precision was δ18O: ± 0.2–0.3‰ and δ13C: ± 0.4–0.5‰. Values of the reference material measurements are listed together with the samples in Supplementary Data 1 (C and O); Supplementary Data 2 (S).

**LA-ICP-MS U-Pb.** U-Pb geochronology via the in situ LA-ICP-MS method was conducted at the Geochronology & Tracers Facility, NERC Isotope Geosciences Laboratory (Nottingham, UK). The method utilizes a New Wave Research 193UC excimer laser ablation system, coupled to a Nu Instruments Attom single-collector sector-field ICP-MS. The method follows that previously described in Roberts et al.[71], and involves a standard-sample bracketing with normalization to NIST 614 silicate glass[72] for Pb-Pb ratios and WC-1 carbonate[71] for U-Pb ratios. The laser parameters comprise a 100 μm static spot, fired at 10 Hz, with a ~8 J/cm$^2$ fluence, for 30 s of ablation. Material is pre-ablated to clean the sample site with 150 μm spots for 3 s. NIST 614 is used for normalization of $^{207}Pb/^{206}Pb$ ratios. No common lead correction is made; ages are determined by regression and the lower intercept on a Tera-Wasserburg plot (using Isoplot 4.15). Duff Brown, a carbonate previously measured by Isotope Dilution mass spectrometry was used as a validation, and pooling of all sessions yields a lower intercept age of 64.2 ± 1.6 Ma (MSWD = 4.0), overlapping the published age of 64.04 ± 0.67 Ma[73]. All ages are plotted and quoted at 2σ and include propagation of systematic uncertainties according to the protocol described in Horstwood et al.[74]. Data were screened for low Pb and low U counts below detection, and very large uncertainties on the Pb-Pb and Pb-U ratios which indicate mixed analyses. The spots are also checked after ablation for consistent ablation pit shape, and data are rejected if the ablations were anomalous (this results from material cleaving off, or clipping the resin mount).

Eight samples of calcite were screened from the Siljan drill cores. Seven samples yielded variably robust U-Pb ages. The other sample did not yield any determinable single age populations that form a regression between common and radiogenic lead compositions. The uranium contents of the samples are variable, as are the initial μ ($^{238}U/^{204}Pb$) values. Along with the requirement of a closed isotopic system, i.e. non-disturbed, the μ values have a large control on the likely success of resolving a precise age, as they dictate the ratio of radiogenic to common lead that may exist in the sample. The samples were measured on one or two occasions, both as small mounted chips, and on larger chips that were previously used for in situ stable isotope analysis. The data do not reflect simple mixing between common and radiogenic lead, but across the grains represent either different age components, variable common lead compositions, and/or disturbed isotopic systematics. The radiogenic data that provide the ages discussed in this manuscript are from pristine

calcite and are interpreted to represent a primary age of this calcite growth. Results are of mixed quality (i.e. both low and high MSWDs), indicating minor open system behavior, and/or mixing between domains of different age for the samples with high MSWD. The interpreted lower intercept ages for the latter are based on radiogenic data and are still useful for broad age interpretation. Full analytical data from the sessions are listed in Supplementary Data 4; ages in Supplementary Data 5; analytical conditions in Supplementary Data 6).

**LA-MC-ICP-MS $^{87}Sr/^{86}Sr$.** The $^{87}Sr/^{86}Sr$ values of the calcite crystals were determined by LA-MC-ICP-MS analysis at the Vegacenter, Swedish Museum of Natural History, Stockholm, using a Nu plasma (II) MC-ICP-MS, and an ESI NWR193 ArF eximer laser ablation system. Ablation frequency was 15 Hz, spot size 80 μm and fluence 2.8 J/cm$^2$ and the same crystal growth zones analyzed with SIMS for δ13C were targeted. Wash-out and ablation times were both 45 s. The $^{87}Sr/^{86}Sr$ analyses were normalized to an in-house brachiopod reference material 'Ecnomiosa gerda' (linear drift and accuracy correction) using a value established by TIMS of 0.709181 (2sd 0.000004[75]). A modern oyster shell from Western Australia was used as a secondary reference material and analyzed at regular intervals together with the primary reference. The accuracy of these analyses was quantified by comparison to the modern seawater value for $^{87}Sr/^{86}Sr$ of 0.7091792 ± 0.0000021[76]. Values of the reference material measurements are listed in Supplementary Data 1.

**GC-MS, biomarkers.** The mineral coating samples (calcite-dominated) were gently flushed with acetone to remove surface contaminations and then ground with an agate pistil. The sample powders were first extracted with 2 mL of pre-distilled dichloromethane/methanol in Teflon-capped glass vials (ultrasonication, 15 min, 40 °C). The supernatant was decanted after centrifuging. Extraction was repeated two times, with dichloromethane and with hexane as solvents. After evaporation of the combined extracts and re-dissolution in pure dichloromethane, the solvents were dried with N$_2$. The total organic extract (TOE) was derivatized by adding 100 μL BSTFA (60 °C, 1.5 h). The sample was dried with N$_2$, and mobilized with 100 μl n-hexane and stored at –18 °C until measurement. The sample remnants were dissolved and demineralized by adding 5 ml of TMCS/Methanol (1 + 9) for 12 h and then derivatised for 90 min at 80 °C. After cooling the samples were mixed with n-hexane and the supernatants were decanted and collected separately. This procedure was repeated three times. The combined supernatants were dried with nitrogen and remobilized with 100 μl n-hexane. 1 μL of each sample extract was analyzed via on-column injection into a Varian CP-3800 GC/1200-quadrupole MS (70 keV) equipped with a fused silica column (Phenomenex ZB-5; 30 m length, 0.32 μm inner diameter; 0.25 μm film thickness). The GC oven was programmed from 80 °C (held 3 min) to 325˚C (held 40 min) at 6 °C/min. He was used as carrier gas at 1.4 ml/min. Compounds were assigned by comparison with published mass spectral data.

**ToF-SIMS.** Right before ToF-SIMS analyzes, the rock containing fractures with $^{13}C$-rich calcite coatings was cracked open, using clean tweezers (heptane, acetone and ethanol in that order), to expose fresh surfaces. This sample was taken from a newly drilled cored borehole. Small pieces of rock containing putative organic remains, and aliquots of dominantly putative organic material, were then mounted with clean tweezers on double-sticky tape on a silica wafer. The ToF-SIMS analysis was performed on a ToF-SIMS IV (ION-TOF GmbH), at RISE, Sweden, by rastering a 25 keV Bi$_3^+$ beam (pulsed current of 0.1 pA) over an area of ~200 × 200 μm for 200–300 s. The analyzes were performed in positive and negative mode at high mass resolution (bunched mode: Δl ~ 3 μm, m∕Δm ~ 2000–4000 at m∕z 30). An electron gun was used for charge composition. As a control, additional spectra were also acquired from the tape to confirm that samples had not been contaminated by the tape.

**Fluid inclusions.** Fluid inclusions were studied using microthermometry techniques for handpicked calcite crystals (0.5–1.5 mm in size). A conventional microscope was used to get an outlook of the samples and the distribution of the fluid inclusions. Microthermometric analyses of fluid inclusions were made with a Linkam THM 600 stage mounted on a Nikon microscope utilizing a ×40 long working-distance objective. The working range of the stage is from −196 to +600 °C. The thermocouple readings were calibrated by means of SynFlinc synthetic fluid inclusions and well-defined natural inclusions in Alpine quartz. The reproducibility was ± 0.1 °C for temperatures below 40 °C and ±0.5 °C for temperatures above 40 °C.

**Gas analysis.** Gases were collected by AB Igrene from boreholes VM2 and VM5 by isotube sampling (and additional isojar sampling of water from the VM2 borehole) and analyzed at commercial laboratory Applied Petroleum Technology AS, Norway. Selected values were extracted from AB Igrene's database for the present work. All laboratory procedures follow NIGOGA, 4th Edition.

Aliquots for GC analysis of gas components of the samples were transferred to exetainers. 0.1–1 ml were sampled using a Gerstel MPS2 autosampler and injected

into a Agilent 7890 RGA GC equipped with Molsieve and Poraplot Q columns and a flame ionization detector.

The $\delta^{13}C$ composition of the hydrocarbon gas components was determined by a GC-C-IRMS system. Aliquots were sampled with a syringe and analyzed on a Trace GC2000, equipped with a Poraplot Q column, connected to a Delta plus XP IRMS. The components were burnt to $CO_2$ and water in a 1000 °C furnace over Cu/Ni/Pt. The water was removed by Nafion membrane separation. Repeated analyses of standards indicate reproducibility better than 1‰ PDB (2 sigma).

The $\delta^{13}C_{CH4}$ values of low $CH_4$ concentrations were determined by a Precon-IRMS system. Aliquots were sampled with a GCPal autosampler. $CO_2$, CO and water were removed on chemical traps. Other hydrocarbons than $CH_4$ and remaining traces of $CO_2$ were removed by cryotrapping. The $CH_4$ was burnt to $CO_2$ and water in a 1000 °C furnace over Cu/Ni/Pt. The water was removed by Nafion membrane separation. The sample preparation system (Precon) was connected to a Delta plus XP IRMS for $\delta^{13}C$ analysis. Repeated analyses of standards indicate reproducibility better than 1‰ PDB (2 sigma).

The $\delta^2H_{CH4}$ isotopic composition of methane was determined by a GC-C-IRMS system. Aliquots were sampled with a GCPal and analyzed on a Trace GC2000, equipped with a Poraplot Q column, connected to a Delta plus XP IRMS. The components were decomposed to $H_2$ and coke in a 1400 °C furnace. The international standard NGS-2 and an in-house standard were used for testing accuracy and precision. Repeated analyses of standards indicate reproducibility better than 10‰ PDB (2 sigma).

## Data availability

All relevant data are included in the Supplementary material to this article. AB Igrene owns the gas data, and for this study selected data have been extracted from their database (placed in Supplementary Data 8).

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

## Acknowledgements
Thanks to AB Igrene for access to drill cores, drilling logs and gas data. Swedish research council (contract 2017–05186 to H.D., 2015–04129 to S.S., 2017–04129 to M.I.) and Formas (contract 2017–00766 to H.D. and M.W.) are thanked for financial support. K. Lindén, M. Tillberg, and M. Schmitt are thanked for analytical or sample preparation assistance, and University of Gothenburg for access to SEM. This is NordSIM publication 617 and Vegacenter publication 20. Open access funding provided by Linnaeus University.

## Author contributions
H.D. initiated and planned the study, carried out sampling, sample preparation, SEM-, SIMS- and LA-MC-ICP-MS analyzes, did the conceptual modeling and wrote the paper. N.R. carried out U-Pb geochronology and data reduction. C.H. carried out biomarker analyzes and interpretation, M.W. handled the SIMS-equipment, instrument tuning and data reduction, S.S. carried out the ToF-SIMS analyzes and data reduction, E.K. handled the LA-MC-ICP-MS-equipment, C.B. carried out fluid inclusion analysis. M.I. and M.Å. were involved in writing.

## Competing interests
The authors declare no competing interests.

## Additional information



