## [Peer Review File · Nature Communications]

Reviewers' comments:

Reviewer #1 (Remarks to the Author):

The Siljan impact structure is the largest-known impact structure in Europe. It has attracted a lot of interest in the 1980s due to assumed occurrences of abiogenic hydrocarbon. The Swedish State Power Board (Vattenfall) started the Deep Gas project in order to evaluate the potential for abiogenic gas production. A few years ago, the Swedish private company AB Igrene has started to explore the area for geothermal energy and natural gas. New boreholes were drilled and cores retrieved from the ring structure. This together with new geophysical surveys provided supplementary information about the subsurface of the ring structure and led to a better understanding of the complex geology of the area. Regardless of these new results, some open questions remain, especially in regard of the origin of the natural gas occurrences.

The study by Drake and co-workers is an interesting contribution toward the ongoing discussion about the source of the hydrocarbons and the mineralization observed in the fractured basement rocks and the Lower Palaeozoic sedimentary cover succession. Drake and co-workers studied core material from several boreholes (named VM1, VM2, CC1, Solberga-1, Stumnsnäs-1, 01-10C, and 01-11C) and from outcrop samples of the Solberga quarry. State-of-the-art chemical and isotopic analyses supplemented by the analysis of organic compounds provide a comprehensive data set. The text is well written and concise. The illustrations are clear and of good quality. The Late Cretaceous to Neogene calcite U-Pb ages are an unexpected, important finding that warrants distribution to the scientific community. Overall, the data set is worth being published in a high-impact journal. I am supportive of this contribution but like to encourage the authors to take following comments into account.

Drake and co-workers did sample various drill cores from the Siljan area but I get the impression that they did not work in detail on the cores themselves. The geology of the area is very complex as one can expect from an impact structure. Therefore, the study by Drake and co-workers largely relies on unpublished reports and published data from the literature. Doing a quick literature research revealed that Arslan et al. (2013) and Lehnert et al. (2013) have first described the Stumnsnäs-1 drill core in detail; I think both papers should be cited accordingly. Arslan et al. (2013) focus on the fractures in the igneous basement rocks and the Lower Palaeozoic cover succession. Mineralization including calcite and oil stain at fractures in limestone are also illustrated and discussed. The references are:

Arslan, A., Meinhold, G. & Lehnert, O., Ordovician sediments sandwiched between Proterozoic basement slivers: tectonic structures in the Stumnsnäs 1 drill core from the Siljan Ring, central Sweden. *GFF* 135, 213-227, doi:10.1080/11035897.2013.809016 (2013).

Lehnert, O., Meinhold, G., Arslan, A., Ebbestad, J. O. R. & Calner, M. Ordovician stratigraphy of the Stumnsnäs 1 drill core from the southern part of the Siljan Ring, central Sweden. *GFF* 135, 204-212, doi:10.1080/11035897.2013.813582 (2013).

Further comments are as follows (The line numbers below refer to the line numbers in the PDF version of the manuscript).

Line 61: I suggest you write "Upper Ordovician"

Line 101: I suggest you write "Upper Ordovician Boda Limestone"

Lines 105-106: You wrote "At Stumnsnäs, the sedimentary succession is sandwiched in between blocks of granite at 196-286 m depth". This exceptional situation has first been described by Arslan et al. (2013, see reference above).

Lines 117-121: Currently, there is no systematic order in referring to the sub-figures of Figure 2. In Line 117 you refer to Fig. 2i and later in Line 119 to Fig. 2a. Please check whether this is okay with the journal guidelines. Commonly, the figures should be referred to in the text, as they appear in the corresponding figure, i.e. in your case Fig. 2a, Fig. 2b, Fig. 2c etc. Please see also the text where you refer to your other figures. For example, in Line 170 you refer to Fig. 6b and in Line 171 to Fig. 6a.

Line 172: I suggest replacing "were yielded" by "were obtained"

Line 209: Insert "is" before "consistent"

Line 231: Write either "Interpretations of the origin of hydrocarbon gases are ..." or "Interpretation of the origin of hydrocarbon gases is ..."

Line 247: Delete "a bit" or replace "a bit" by "slightly"

Line 308: Replace "yielded" by "obtained"

Line 405: Write "... is of Late Neoproterozoic-Early Paleozoic age (Fig. S1), ..."

Line 409: Insert "Late" before "Neoproterozoic"

Line 454: Replace "indicates" by "indicate"

Lines 682-879: Carefully check your references for full citation and consistency.

Figure 1: Use italic font for the water bodies. Use bold font for the sample locations, i.e. the borehole sites and the Solberga quarry. It seems that some of the layers in your map did shift to the left, e.g. the road and the inset map appear to be slightly out of place.

Supplementary material

In Supplementary Table S1, in drill core VM2 you mention white limestone at 177 m, 212 m and 328 m depths. Of what age is that limestone? To which stratigraphic member/formation does it belong?

In Supplementary Table S1, in drill core CC1 you mention grey limestone at 64 m depth and grey limestone with clayey parts at 402.6 m, 405 m and 405.8 m depths. Of what age is that limestone? To which stratigraphic member/formation does it belong?

In Supplementary Table S1, in drill core 01-10C you mention white limestone with oil seep at 324 m depth and grey limestone with clayey parts at 345 m, 351 m, 353 m and 356 m depths. Of what age is that limestone? To which stratigraphic member/formation does it belong?

Supplementary Table S4 shows the U-Pb isotope data. I suggest you explain the meaning of the color coding (black, blue, red, and green). Furthermore, I suggest you insert a new column where you indicate with an asterisk which spot analyses have been used for average age calculation. Currently, it is very difficult to relate the values given in Table S4 to those in Table S5.

In the caption of the Supplementary Figure S1, in the first sentence, it says "... that failed to results in a U-Pb age solution (c,d)." You show a Figure (c) but Figure (d) is missing.

In the caption of the Supplementary Figure S3 check the usage of commas and full stops.

Reviewer #2 (Remarks to the Author):

- What are the major claims of the paper?

The paper attempts to elucidate the source(s) and timing of methane production within the Devonian Siljan ring in Sweden. I generally agree with the authors' contention that the strongly isotopically enriched carbonates suggest the occurrence of microbial methanogenesis within the system, and that this occurred more recently than the structure-forming impact. This is supported by observation of isotopically depleted methane $\delta^{13}\text{C}$ values, and methane plotting close to the microbial fields in Figure 9. I also agree that the simplest explanation for the highly isotopically depleted carbonates would be oxidation of isotopically light methane by a process such as anaerobic methane oxidation. However, this process would be expected to also result in isotopic enrichment of the residual methane, as argued for the carbonate system. Overall, Figure 9 is not unequivocal given that many of the points plot within multiple fields and show evidence of mixed sources. I feel that the carbonate isotope evidence does support the authors' contention. While I am not an expert in the dating approaches used, the evidence of relatively recent ages (10's of millions of years) for these calcites supports a geologically recent origin for both of these types of carbonates ($\delta^{13}\text{C}$ enriched and depleted). However, I have some questions (see below) that I feel the authors have not sufficiently addressed regarding their proposed model for the system and their interpretations of some aspects of the data. I think that the discussion of these points would need to be addressed prior to the publication of this paper.

I think that the authors need to do a better job of demonstrating the novelty and impact of this study. It is of interest that the authors are revisiting a geological locale that has been of historical interest. However, the authors allude to their model having implications to other environments, but don't really explain what those implications are or their significance. This would need to be more clearly expressed to convince the reader of the novelty of the work. The authors have tackled the challenging problem of differentiating sources of methane in complex systems. The use of SIMS and high spatial resolution analysis of the isotopic compositions of carbonates and their dating in this context is novel to my knowledge. However, the overall mechanisms and signatures being assessed are derived from traditional approaches.

There are several aspects of the authors' interpretations that I feel need to be revisited prior to this paper being acceptable for publication.

I do not agree with the authors' contention that there is evidence that biodegradation of hydrocarbons by sulfate-reducing organisms is the most likely source for dissolved inorganic carbon in the system. The sedimentary rocks in question are limestone. While no aqueous data is presented, I think it reasonable that the waters within the system would be in equilibrium with the rock and thus saturated with DIC. The authors would need to demonstrate that biodegradation would produce significant concentrations of DIC compared to the equilibrium of the rocks within the system to support their point. It should also be noted that production of the DIC from respiration of organic matter would produce isotopically depleted DIC, requiring even further enrichment than carbonate-derived DIC. Further, the analysis of organic carbon present in the system presented does not show strong evidence of biodegradation. The data shows well-resolved alkanes with little development of an unresolved complex mixture generally associated with biodegradation. While biodegradation may certainly have happened, it does not appear to have been a dominant effect. However, biodegradation, specifically fermentation, may supply the H_2 that would be required by the microbial communities to carry out CO_2 reduction methanogenesis. The authors do not address this aspect of the system at all in their current model.

The authors also do not really address the implications of the co-variation in Sr ratios with the variation in calcite $\delta^{13}\text{C}$ values. They note a constant source for the older carbonates, but do not effectively discuss where distinct values associated with the enriched carbonates are coming from, nor what the implication is to the variation in the carbonate values.

I did not find that the analysis of the "PAHs" in the organic residue added strong support to the authors' arguments. Some of the masses were consistent with PAH structures, but many of the stated formulae were not consistent with PAH structures. Further, it is not clear to me that this

supports the occurrence of biodegradation. And I think there is other, strong demonstration of the presence of organics within the system.

In addition to these general, conceptual points, the authors also need to address the following points:

Line 48 Define the zone being considered the aquifer

Line 51 It should be specifically stated that the sedimentary rocks are carbonates, the presence of carbonate matrix rocks needs to be dealt with in the discussion.

Line 218 I had to dig into the excel file to find the concentration of CO₂ in the gas, this data needs to be addressed to allow the reader to assess the extent of methane generation or oxidation required to generate the observed isotopic shifts.

Line 221 I agree there is spatiotemporal variation

Line 223 This syntax is confusing "ancient processes producing the youngest calcite", perhaps just say processes producing the youngest calcite

Line 233 I believe the authors mean to refer to Figure 9

Line 248 Yes there could be a mixed origin, however, it is difficult to interpret the data in Figure 9 considering that it is relatively limited and not clear how it relates to the enriched calcites spatially. The data in Figure 9 plots either within multiple zones, or at the borders of these empirically defined zones on the plots, which is not strongly convincing of source.

Line 267 The argument regarding the ethane and propane is plausible, however it does not address the lack of evidence of degradation of the butane in the same sample. Further this is only one sample from a complex system. The authors would be able to make much more effective arguments with access to further data if that can be obtained from the industrial owners of the data.

Line 300 I agree microbial methane appears to be present

Line 307 I am not convinced that degraded hydrocarbons are the source of the DIC for methane production

Line 382 I agree that there is evidence of sulfur cycling within the system, however not that it can be related to production of the CO₂. I can accept that there may be evidence of anaerobic methane oxidation generating the observed $\delta^{13}\text{C}$ depleted carbonates.

Line 426 I am not convinced of the conceptual model presented. I think the potential sources of DIC related to the carbonates need to be dealt with. I do not see strong evidence of biodegradation of the organics. But I do agree there appears to be evidence of microbial methanogenesis and methanotrophy recorded in the carbonates.

Line 477 I am not convinced that the microbial methanogenesis has produced large quantities of methane

- Will the paper be of interest to others in the field?

This is an interesting system to address, however, the authors need to be more convincing of the explanation of what is happening and the breadth of its implications

- Will the paper influence thinking in the field?

I am not sure how much this will influence thinking in the field. It is an interesting system, but I am not convinced it is highly unique by the data and arguments presented.

- Are the claims convincing? If not, what further evidence is needed?

See comments above

- Are there other experiments that would strengthen the paper further? How much would they improve it, and how difficult are they likely to be?

Further presentation of water and gas chemistry and isotopes would potentially fill out the argument. But I recognize that this data may not be easily obtained.

- Are the claims appropriately discussed in the context of previous literature?

Yes

- If the manuscript is unacceptable in its present form, does the study seem sufficiently promising that the authors should be encouraged to consider a resubmission in the future?

I was not convinced that the authors have demonstrated the novelty and significance of the story sufficiently.

-

Is the manuscript clearly written? If not, how could it be made more accessible?

The manuscript is well written

- Could the manuscript be shortened to aid communication of the most important findings?

There are some points, like the analysis of the organic residue by SIMS, that do not clearly currently contribute to the story.

- Have the authors done themselves justice without overselling their claims?

The claims of the paper need to be more clearly presented.

- Have they been fair in their treatment of previous literature?

Yes

- Have they provided sufficient methodological detail that the experiments could be reproduced?

Yes

- Is the statistical analysis of the data sound?

Not a key component

- Should the authors be asked to provide further data or methodological information to help others replicate their work? (Such data might include source code for modelling studies, detailed protocols or mathematical derivations).

It was difficult to find some of the supporting material that was located in the excel file. Some of this data might warrant inclusion in the paper.

- Are there any special ethical concerns arising from the use of animals or human subjects?

No

Reviewer#1

The Siljan impact structure is the largest-known impact structure in Europe. It has attracted a lot of interest in the 1980s due to assumed occurrences of abiogenic hydrocarbon. The Swedish State Power Board (Vattenfall) started the Deep Gas project in order to evaluate the potential for abiogenic gas production. A few years ago, the Swedish private company AB Igrene has started to explore the area for geothermal energy and natural gas. New boreholes were drilled and cores retrieved from the ring structure. This together with new geophysical surveys provided supplementary information about the subsurface of the ring structure and led to a better understanding of the complex geology of the area. Regardless of these new results, some open questions remain, especially in regard of the origin of the natural gas occurrences.

The study by Drake and co-workers is an interesting contribution toward the ongoing discussion about the source of the hydrocarbons and the mineralization observed in the fractured basement rocks and the Lower Palaeozoic sedimentary cover succession. Drake and co-workers studied core material from several boreholes (named VM1, VM2, CC1, Solberga-1, Stumnsås-1, 01-10C, and 01-11C) and from outcrop samples of the Solberga quarry. State-of-the-art chemical and isotopic analyses supplemented by the analysis of organic compounds provide a comprehensive data set. The text is well written and concise. The illustrations are clear and of good quality. The Late Cretaceous to Neogene calcite U-Pb ages are an unexpected, important finding that warrants distribution to the scientific community. Overall, the data set is worth being published in a high-impact journal. I am supportive of this contribution but like to encourage the authors to take following comments into account.

Response: We acknowledge this positive and constructive assessment. We have answered and taken action to all comments, as detailed below.

Drake and co-workers did sample various drill cores from the Siljan area but I get the impression that they did not work in detail on the cores themselves. The geology of the area is very complex as one can expect from an impact structure. Therefore, the study by Drake and co-workers largely relies on unpublished reports and published data from the literature. Doing a quick literature research revealed that Arslan et al. (2013) and Lehnert et al. (2013) have first described the Stumnsås-1 drill core in detail; I think both papers should be cited accordingly. Arslan et al. (2013) focus on the fractures in the igneous basement rocks and the Lower Palaeozoic cover succession. Mineralization including calcite and oil stain at fractures in limestone are also illustrated and discussed. The references are:

Arslan, A., Meinhold, G. & Lehnert, O., Ordovician sediments sandwiched between Proterozoic basement slivers: tectonic structures in the Stumnsås 1 drill core from the Siljan Ring, central Sweden. *GFF* 135, 213-227, doi:10.1080/11035897.2013.809016 (2013).

Lehnert, O., Meinhold, G., Arslan, A., Ebbestad, J. O. R. & Calner, M. Ordovician stratigraphy of the Stumnsås 1 drill core from the southern part of the Siljan Ring, central Sweden. *GFF* 135, 204-212, doi:10.1080/11035897.2013.813582 (2013).

Response: It is true that we did not do detailed geological descriptions of the whole cores. Instead, we focused on the sections with observed gas and oil seep and

relied on the excellent previous core descriptions of Lehnert and colleagues in particular. We have used the above mentioned references in our study but did unfortunately not cite them in the previous submission due to 1) journal limitation of references 2) Stumnsnäs-1 can be considered to be the least interesting core from a methane perspective as there was no significant gas flow detected during drilling. Thus we only took two samples from this hole. However, we have now added the citations and related mineral descriptions as requested by the reviewer and we agree it is important to acknowledge these previous works, accordingly: “*At Stumnsnäs, the sedimentary succession is sandwiched in between blocks of granite at 196-286 m depth and oil stains and calcite mineralizations have been observed in limestone fractures* (suggested references inserted).”

Further comments are as follows (The line numbers below refer to the line numbers in the PDF version of the manuscript).

Line 61: I suggest you write “Upper Ordovician”

Line 101: I suggest you write “Upper Ordovician Boda Limestone”

Response: Both occurrences above corrected.

Lines 105-106: You wrote “At Stumnsnäs, the sedimentary succession is sandwiched in between blocks of granite at 196-286 m depth”. This exceptional situation has first been described by Arslan et al. (2013, see reference above).

Response: See answer above.

Lines 117-121: Currently, there is no systematic order in referring to the sub-figures of Figure 2. In Line 117 you refer to Fig. 2i and later in Line 119 to Fig. 2a. Please check whether this is okay with the journal guidelines. Commonly, the figures should be referred to in the text, as they appear in the corresponding figure, i.e. in your case Fig. 2a, Fig. 2b, Fig. 2c etc. Please see also the text where you refer to your other figures. For example, in Line 170 you refer to Fig. 6b and in Line 171 to Fig. 6a.

Response: This is generally a correct remark but can be hard to do properly when using a complex multi-panel figure such as figure 2 showing photos, SEM-images and transects. We have, however, moved most text descriptions, so that they appear in correct order.

Line 172: I suggest replacing “were yielded” by “were obtained”

Response: Corrected as suggested.

Line 209: Insert “is” before “consistent”

Response: Corrected as suggested.

Line 231: Write either “Interpretations of the origin of hydrocarbon gases are ...” or “Interpretation of the origin of hydrocarbon gases is ...”

Response: Corrected as suggested (to “Interpretation...”).

Line 247: Delete “a bit” or replace “a bit” by “slightly”

Response: Corrected as suggested (to “slightly”).

Line 308: Replace “yielded” by “obtained”

Response: Corrected as suggested.

Line 405: Write “... is of Late Neoproterozoic-Early Paleozoic age (Fig. S1), ...”

Response: Corrected as suggested.

Line 409: Insert “Late” before “Neoproterozoic”

Response: Corrected as suggested.

Line 454: Replace “indicates” by “indicate”

Response: Corrected as suggested.

Lines 682-879: Carefully check your references for full citation and consistency.

Response: Checked and corrected.

Figure 1: Use italic font for the water bodies. Use bold font for the sample locations, i.e. the borehole sites and the Solberga quarry. It seems that some of the layers in your map did shift to the left, e.g. the road and the inset map appear to be slightly out of place.

Response: Corrected as suggested.

Supplementary material

In Supplementary Table S1, in drill core VM2 you mention white limestone at 177 m, 212 m and 328 m depths. Of what age is that limestone? To which stratigraphic member/formation does it belong?

In Supplementary Table S1, in drill core CC1 you mention grey limestone at 64 m

depth and grey limestone with clayey parts at 402.6 m, 405 m and 405.8 m depths. Of what age is that limestone? To which stratigraphic member/formation does it belong?

In Supplementary Table S1, in drill core 01-10C you mention white limestone with oil seep at 324 m depth and grey limestone with clayey parts at 345 m, 351 m, 353 m and 356 m depths. Of what age is that limestone? To which stratigraphic member/formation does it belong?

Response: As stated above, we did not do a full geological description of the drill cores and stratigraphic units, but focused entirely on the secondary coatings, gases and organic remains. Interpretations of the units require a thorough stratigraphic study, such as the excellent ones of Lehnert and colleagues have done on the other cores. We encourage future work to provide more details on the stratigraphy of these cores, but so far, to our knowledge, no detailed work has yet been done on these (just very simple mapping by the prospecting company stating e.g. white/red limestone/granite/shale etc). We have now written this in the footnote of this table: “Stratigraphic information is retrieved from Lehnert et al., (2012, 2013) and Arslan et al., (2013). More general descriptions (e.g. "white limestone") are given for cores that have not yet been thoroughly stratigraphically documented.”

Supplementary Table S4 shows the U-Pb isotope data. I suggest you explain the meaning of the color coding (black, blue, red, and green). Furthermore, I suggest you insert a new column where you indicate with an asterisk which spot analyses have been used for average age calculation. Currently, it is very difficult to relate the values given in Table S4 to those in Table S5.

Response: The meaning of color coding is now explained in the footnote “Color coding refers to different rock chips and/or crystals used for age determination (see corresponding colors in Fig. 6 and Fig. S1)”. The same colors have been used as coding in Table S5 for clarity.

In the caption of the Supplementary Figure S1, in the first sentence, it says "... that failed to results in a U-Pb age solution (c,d)." You show a Figure (c) but Figure (d) is missing.

Response: That part of the sentence was deleted as the details to (c) already tells that there was no age solution for this particular sample. The repetition of “no age solution” was deleted as well.

In the caption of the Supplementary Figure S3 check the usage of commas and full stops.

Response: Checked and corrected.

Reviewer#2

The paper attempts to elucidate the source(s) and timing of methane production within the Devonian Siljan ring in Sweden. I generally agree with the authors' contention that the strongly isotopically enriched carbonates suggest the occurrence of microbial methanogenesis within the system, and that this occurred more recently than the structure-forming impact. This is supported by observation of isotopically depleted methane $\delta^{13}\text{C}$ values, and methane plotting close to the microbial fields in Figure 9. I also agree that the simplest explanation for the highly isotopically depleted carbonates would be oxidation of isotopically light methane by a process such as anaerobic methane oxidation. However, this process would be expected to also result in isotopic enrichment of the residual methane, as argued for the carbonate system. Overall, Figure 9 is not unequivocal given that many of the points plot within multiple fields and show evidence of mixed sources. I feel that the carbonate isotope evidence does support the authors' contention. While I am not an expert in the dating approaches used, the evidence of relatively recent ages (10's of millions of years) for these calcites supports a geologically recent origin for both of these types of carbonates ($\delta^{13}\text{C}$ enriched and depleted). However, I have some questions (see below) that I feel the authors have not sufficiently addressed regarding their proposed model for the system and their interpretations of some aspects of the data. I think that the discussion of these points would need to be addressed prior to the publication of this paper.

Response: We acknowledge the overall positive assessment on the mineral interpretations and address the remarks below.

I think that the authors need to do a better job of demonstrating the novelty and impact of this study. It is of interest that the authors are revisiting a geological locale that has been of historical interest. However, the authors allude to their model having implications to other environments, but don't really explain what those implications are or their significance. This would need to be more clearly expressed to convince the reader of the novelty of the work. The authors have tackled the challenging problem of differentiating sources of methane in complex systems. The use of SIMS and high spatial resolution analysis of the isotopic compositions of carbonates and their dating in this context is novel to my knowledge. However, the overall mechanisms and signatures being assessed are derived from traditional approaches.

Response: We are pleased to see that the reviewer recognizes the novelty of the used techniques and the challenges of interpreting sources of methane in this complex system. This work presents the first detailed evidence for long-term methane formation in this famous meteorite structure, which should be of large scientific interest in itself. However, we realize now that the significance must go beyond understanding this structure, and this can indeed be done more carefully. We have therefore made a thorough revision of the implications and their significance, and we are grateful to the reviewer for pointing it out, as it greatly strengthens the manuscript. This is the most

extensive study of long-term microbial processes in general, and of complex methane cycling in particular, in a major terrestrial impact structure. The implications of this work should, in contrast to what we previously expressed in the manuscript, be seen more in the context of terrestrial impact structures, and the new understanding of them as deep microbial habitats with influence on methane cycling. The understanding of deep life in these structures also has astrobiological implications that we now address in the revised manuscript. Our findings confirm that impact craters are favorable microbial habitats on Earth. We also discuss challenges of using impact structures as astrobiological analogs and show that extensive multi-disciplinary micro-scale studies are required to distinguish between impact-related and post-impact colonization. The methodology presented should therefore be optimal to provide spatiotemporal constraints for ancient microbial methane formation and consumption in other impact crater systems, such as the methane emitting craters on Mars. We also discuss the recognition of deep microbial methanogenesis in the upper crystalline crust, that may be widespread in space and time (this part was covered a bit previously but we have stressed why it is important). Changes appear at several occasions, but key additions are in the updated abstract, introduction (first and last paragraphs, lines 28-41, and 101-110) and discussion (last two paragraphs, lines 517-561). These additions/updates are extensive and we decided not to paste them in full into this response.

There are several aspects of the authors interpretations that I feel need to be revisited prior to this paper being acceptable for publication.

I do not agree with the authors contention that there is evidence that biodegradation of hydrocarbons by sulfate reducing organisms is the most likely source for dissolved inorganic carbon in the system. The sedimentary rocks in questions are limestone. While not aqueous data is presented, I think it reasonable that the waters within the system would be in equilibrium with the rock and thus saturated with DIC. The authors would need to demonstrate that biodegradation would produce significant concentrations of DIC compared to the equilibrium of the rocks within the system to support their point. It should also be noted that production of the DIC from respiration of organic matter would produce isotopically depleted DIC, requiring even further enrichment than carbonate derived DIC. Further, the analysis of organic carbon present in the system presented do not show strong evidence of biodegradation. The data shows well resolved alkanes with little development of an unresolved complex mixture generally associated with biodegradation. While biodegradation may certainly have happened it does not appear to have been a dominant effect. However, biodegradation, specifically fermentation may supply the H₂ that would be required by the microbial communities to carry out CO₂ reduction methanogenesis. The authors do not address this aspect of the system at all in their current model.

Response: The reviewer is correct about our pathway model from organic matter to methane that focused too much on the oxidation of organic matter via microbial sulfate reduction. This was based on the observed S isotope signatures in pyrite. It is likely more complex and involves fermentation producing H₂ that can be used by methanogens for CO₂ reduction. It is a very good suggestion to add fermentation pathway in the conceptual model, and

this would fit with how the values plot in the discrimination diagram (Fig. 8); on the borders between fermentation, carbonate reduction and thermogenic. We acknowledge this input and have presented a new model that involves fermentation of organic matter and potential influence of DIC (see below). The heavy $\delta^{13}\text{C}_{\text{CO}_2}$ and light $\delta^{13}\text{C}_{\text{CH}_4}$ values still suggest that carbonate reduction have been important as the final methanogenesis step, in accordance with studies of biodegradation in petroleum reservoirs.

Regarding the influence on DIC from limestone: The reviewer is right that limestone is an important lithology in the system. The influence on DIC may be an important process in the carbonate rock without bitumen and seep oil, but we focus on fractures with documented bitumen, seep oil, and gas. It is important to note 1) a large part of the system is in the granitic basement rock (>200 m below the contact), this means that DIC is not controlled by limestones in all parts of the system, 2) similar $^{13}\text{C}_{\text{calcite}}$ enrichment occurs in calcite in limestone and granite, and 3) Bitumen and seep oils of black shale origin are dispersed in the rock fractures and also in the limestone matrix pore space. We regret that the reviewer misinterpreted this from our discussion, which generally described the aquifer to be of limestone lithology for simplicity and we have clarified this now when describing the limestone-shale-granite system and the dispersed oil and bitumen occurrences. We have also added available water chemistry data of the deep aquifer to support our interpretations, added $\delta^{13}\text{C}$ values for the limestone and elaborated more on these parts in the discussion, as well as updated the conceptual model. When microbial utilization of substrates occurs locally in the fracture system its' products are probably, as the reviewer point out, diluted by the isotopic signature of limestone derived DIC. However, microbial processes are kinetic and can influence the isotope pool in microscale of a stagnant fracture system aquifer. Microbial carbon utilization has thus influenced the DIC signature locally in the system (close to where methanogenesis occurs in various parts of the fracture system). In our previous studies we have observed very different and varied isotope signatures in minerals compared to the bulk isotope signatures of the water the minerals precipitated from. It is thus fully possible to produce large isotopic variability in microscale close to biofilms in systems that can be expected to be buffered by DIC. Nevertheless, influence (dilution) of limestone-derived DIC can be important as the reviewer point out to provide CO_2 for microbial methanogenesis through carbonate reduction and would make it easier to produce the observed heavy $\delta^{13}\text{C}_{\text{calcite}}$ during methanogenesis. Utilization by methanogens of CO_2 from the limestone for reduction would require an electron donor, such as H_2 formed during fermentation, as the reviewer pointed out above. We have now broadened the discussion about the potential methanogenesis pathways and the influence of DIC according to the reviewers suggestions, accordingly: "The limestones in VM-1 and Solberga-1 have $\delta^{13}\text{C}$ values of 0 to +2‰⁵² and shallow wells in the aquifer contain HCO_3^- concentrations of up to 400 mg/L, compared to 10-80 mg/L in the granitic rock aquifer⁵⁰. In the limestone aquifer, it is more likely that equilibrium between DIC and the wall rock dominates. Methanogenesis through carbonate reduction of limestone derived DIC would require smaller ^{13}C enrichment than utilization of ^{13}C -poor DIC formed by oxidation of organic matter to reach the heavy $\delta^{13}\text{C}_{\text{calcite}}$ values observed. This means that

microbial carbonate reduction may to some degree have utilized limestone derived DIC, at least in the limestone aquifer. However, a system with abundant DIC would likely dilute and mask methanogenesis-related $\delta^{13}\text{C}$ signatures in the produced carbonates. Nevertheless, there is evidently substantial $^{13}\text{C}_{\text{calcite}}$ and $^{13}\text{C}_{\text{CO}_2}$ enrichment in the sedimentary aquifer. It has been shown in deep energy-poor fracture systems that isotopic fractionation and distillation can occur in microscale in biofilms resulting in isotopic compositions of produced minerals that are very different from the bulk groundwater. At Äspö in Sweden, pyrite precipitated over a 17yr period from a deep sulfate-rich water with relatively constant $\delta^{34}\text{S}_{\text{sulfate}}$ of 20-30‰ had $\delta^{34}\text{S}$ values of -47.3 to +53.3‰⁵³ and at nearby sites, calcite had extreme $\delta^{13}\text{C}$ variation (-125 to +37‰) compared to the corresponding deep groundwater $\delta^{13}\text{C}_{\text{DIC}}$ (-17±3‰)²⁷. We propose that similar kinetic microbial processes have locally influenced the DIC signature in the Siljan aquifer, particularly in pore space infiltrated by gases, bitumen and seep oils, as shown by spatial relation of these features to significantly ^{13}C -rich calcite (Fig. 3).” At lines 361-380. We also expanded the section on fermentation, and downgraded the importance of MSR, and provided geochemical data for sulfate and chlorine: “Methanogenesis is commonly associated with sulfate poor biodegraded petroleum reservoirs⁴⁸ and initial steps of anaerobic utilization of organic matter (fermentation) involve hydrolysis followed by bacterial acetogenesis that converts volatile fatty acids into acetic acid, H_2 and CO_2 ⁵⁴. Alternatively, H_2 is produced by aromatization of compounds present in the seep oil⁴⁸. Methanogenesis through CO_2 reduction has been proposed to be the dominant terminal process in petroleum biodegradation in the subsurface⁵⁵, and this appears also to be the case at Siljan based on the widespread and pronounced heavy $\delta^{13}\text{C}_{\text{calcite}}$ and $\delta^{13}\text{C}_{\text{CO}_2}$ values (Figs. 4 and 8c). The supply of the electron donor H_2 needed for microbial methanogenesis through CO_2 reduction may thus originate from fermentation of hydrocarbon related organic matter, in addition to geological sources (hydrolysis). This is corroborated by positive correlation of H_2 and C_1 in head space gas from the granitic aquifer⁵⁰. In sulfate rich reservoirs, microbial sulfate reduction (MSR) can be involved in degradation of hydrocarbons. In the Siljan fractures, pyrite occasionally occurs together with ^{13}C -enriched calcite. Pyrite formed by MSR is typically strongly depleted in ^{34}S (ref. ⁵⁶). The very low minimum $\delta^{34}\text{S}_{\text{pyrite}}$ values (-40‰ V-CDT, Fig. 5) is thus proposed to reflect MSR. However, groundwater in granite of adjacent boreholes show very low sulfate concentrations, 4.3 and 6.6 mg/L at ~180 and 460 m, respectively⁵⁰, suggesting a generally low potential for MSR in that aquifer. Although anaerobic oxidation of organic matter by MSR can produce CO_2 that can be utilized by methanogens⁴⁶, it did probably not result in large quantities of methane because sulfate reducers outcompete methanogens for H_2 and other substrates when sulfate concentrations are elevated⁵⁷. Instead, fermentation likely dominated initial degradation steps of organics in the system providing H_2 for the methanogens to perform reduction of CO_2 formed by e.g. fermentation and/or being limestone-derived, as discussed above. Furthermore, low salinity (Cl^- : 1-2 mg/L⁵⁰) in the deep granite aquifer is favorable for microbial methanogenesis, which is suppressed at high salinities⁵⁸.” At lines 383-407.

Consequently, the conceptual model was updated accordingly: “The exceptional ^{13}C -enrichment in calcite and spatially related biodegraded bitumen and seep oil suggest secondary methane formation following anaerobic degradation of organic matter through e.g. fermentation that produces H_2 for utilization by methanogens through reduction of CO_2 formed during biodegradation or occurring in the aquifer (in limestone mainly). The kinetic microbial processes producing microbial methane resulted in large isotopic fractionations, as observed in the gases and secondary carbonates.”

Response: Regarding biodegradation. We agree that we could have shown more proofs for biodegradation, because there are definitely more support to prove our point in our dataset and in previous works. We have made a comprehensive effort to present evidence of biodegradation as robustly as possible. We do this by adding 1) more details to the discussion of the biomarkers in the calcite coatings (organic molecules, like the *n*-alkane pattern of the granite fractures indicative for moderate to severe biodegradation in combination with the distinct fatty acids characteristic for the deep microbial activity *in situ*, 2) discussion of Siljan seep oil and bitumen data presented earlier (Ahmed et al., 2014), which show very clear signs of biodegradation (including large hump of unresolved complex mixture of hydrocarbons in seep oil and bitumen and preferential removal of almost all the *n*-alkanes and alkylcyclohexanes in seep oil), 3) more gas data and focusing the discussion more towards biodegradation signatures in the primary gas (including evidence of higher utilization of *n*-alkanes than *i*-alkanes suggesting microbial utilization of primary thermogenic gas), and, We have now expanded the discussion of this section. Major additions for 1-3 above are:

For 1 and 2 combined, at lines 332-356: “In addition, the preserved fatty acids *n*- C_{12} to *n*- C_{18} , particularly the odd chain and branched fatty acids *i* C_{15} , *ai* C_{15} , *n*- C_{15} , 12Me- C_{16} , *ai* C_{17} , and 12OH- C_{18} as well as the *n*-alcohols and the 1-*o*-*n*-hexadecylglycerol preserved within ^{13}C -rich calcite coatings are support for *in situ* microbial activity. Preserved hydrocarbon *n*-alkane pattern of calcite in bitumen-bearing fractures of the sedimentary rock and at the sediment-granite interface (VM2:212; VM1:251, Fig. 7) is indication for thermal- and biodegradation (black shale), and it has previously been reported that biomarkers in bitumen in sedimentary rock fractures link its origin to shales²¹. Bimodal *n*-alkane distribution indicates that the Solberga bitumen formed by mixing of more than one charge of oil at various degrees of degradation²¹ suggesting mobilization and degradation of hydrocarbons at several events in the fracture system. Furthermore, presence of a large hump of unresolved complex mixture (UCM) of hydrocarbons in seep oil and bitumen and preferential removal of almost all the *n*-alkanes and alkylcyclohexanes in seep oil are indicative of moderate to severe biodegradation of these materials in the limestone²¹. The irregular *n*-alkane distribution in combination with humps of UCM in the granite fractures (CC1:539; CC1:608) indicate more severe biodegradation of organic matter (Fig. 7). The hydrocarbon distribution with poor straight chain carbons and S&R-hopanoid isomers in the ^{13}C -rich calcite coatings in the granite fractures indicate moderate to severe biodegradation. Although the sedimentary rocks and the deeper granite fractures contain widespread bitumen and seep oil stains, these carbon sources were in the investigated

samples predominantly used by the microbial communities in the granite fractures, likely due to the lack of other substrates (Fig. 7). This is in line with calculated carbon preference index values indicate influence for migrated seep oil/bitumen similar to Bitumen 1 [ref²¹], also deep within the granite (Fig. 7, Table S7). These levels of degradation are corroborated by the very small amounts of C₂₉ 25-norhopane in the seep oils and bitumen²¹.”

3: at lines 297-311: “Biodegradation signatures of the hydrocarbon gas include high C₂ to C₃ ratios owing to that ethane is relatively resistant to biodegradation compared to the C₃₊ homologues⁴⁷. Biodegradation also discriminates against ¹³C_{C3}, leading to isotopically heavy residual propane⁴⁷. The anomalously heavy Siljan gas δ¹³C_{C3} values compared to the δ¹³C_{C2} values, as well as the high C₂ to C₃ ratios, which are far from the normal range for thermogenic gases thus suggest biodegradation. The lack of ¹³C-enrichment in C₄ compared to C₃ can be due to that propane is biodegraded most rapidly of the hydrocarbons, already at slight levels of biodegradation⁴⁸. *n*-alkanes are preferentially utilized during biodegradation of oil and gas⁴⁶. The high *i*-C₄/*n*-C₄ in the Siljan gas (Table S8, as well as higher *i*-C₅ than *n*-C₅ [below detection]) also suggests significant microbial utilization of primary thermogenic gas, and analogously does high *neo*-C₅/*i*-C₅, because *neo*-C₅ is relatively resistant to biodegradation⁴⁹. There are thus several lines of evidence in support of biodegradation of the thermogenic gas fraction, but the degree of degradation is difficult to assess. The removal of the higher hydrocarbons during biodegradation increases the C₁/(C₂+C₃), which complicates the estimation of the mixing proportions between microbial and thermogenic gas.”

The authors also do not really address the implications of the co-variation in Sr ratios with the variation in calcite δ¹³C values. They note a constant source for the older carbonates, but do not effectively discuss where distinct values associated with the enriched carbonates are coming from, nor what the implication is to the variation in the carbonate values.

Response: The increase in ⁸⁷Sr/⁸⁶Sr with time is due to beta decay of ⁸⁷Rb, and this increase is in line with the distinct difference in U-Pb ages of the two major calcite populations. We added: “The significantly higher ⁸⁷Sr/⁸⁶Sr values of the Late Cretaceous to Neogene calcite compared to the Late Neoproterozoic-Early Paleozoic calcite (Fig. 2) are proposed to result from increased ⁸⁷Sr with time due to beta decay of ⁸⁷Rb in the rocks, and prolonged water rock interaction. This increase is, as expected, of largest magnitude in calcite lining Rb-rich granitic wall rock. The higher ⁸⁷Sr/⁸⁶Sr of the calcite overgrowths is a relative timing indicator that is in agreement with the U-Pb dating of two temporally separated calcite populations.” To lines 449-455.

I did not find that the analysis of the “PAHs” in the organic residue added strong support to the authors arguments. Some of the masses were consistent

with PAH structures, but many of the stated formulae were not consistent with PAH structures. Further it is not clear to me that this supports the occurrence of biodegradation. And I think there is other, strong demonstration of the presence of organics within the system.

Response: Although the PAH signature is in accordance with previously reported PAH from seep oil in the area, we generally agree that there are stronger demonstrations of the presence of organics and biodegradation. We have therefore moved most of the ToF-SIMS text and the figure to the supplementary and left room to focus more on other organic signatures.

In addition to these general, conceptual points, the authors also need to address the following points:

Line 48 Define the zone being considered the aquifer

Response: "...the Siljan crystalline and sedimentary rock aquifers is yet to be proven"

Line 51 It should be specifically stated that the sedimentary rocks are carbonates, the presence of carbonate matrix rocks needs to be dealt with in the discussion.

Response: The sedimentary rocks are not only limestones. Most calcite samples in the sedimentary rocks are from limestone but shales (including black shale) is also important. For instance in the VM-1 core, 190 of 240 m sedimentary rock core length is shale. We added here: "...thick¹⁹ down-faulted Ordovician and Silurian sedimentary rocks (dominantly limestone but shales are also abundant)" on lines 70-71.

Line 218 I had to dig into the excel file to find the concentration of CO₂ in the gas, this data needs to be addressed to allow the reader to assess the extent of methane generation or oxidation required to generate the observed isotopic shifts.

Response: Good point. We added these concentrations to the main text: "...dominated by methane (mainly >90%, whereas CO₂ is at 3-14% in the gas samples, with even higher CO₂ in the water samples)." at lines 233-234.

Line 221 I agree there is spatiotemporal variation

Line 223 This syntax is confusing "ancient processes producing the youngest calcite", perhaps just say processes producing the youngest calcite

Response: Good suggestion. We changed according to this suggestion.

Line 233 I believe the authors mean to refer to Figure 9.

Response: True. Now changed to Figure 8 anyway as we moved ToF-SIMS figure to supplementary.

Line 248 Yes there could be a mixed origin, however, it is difficult to interpret the data in Figure 9 considering that it is relatively limited and not clear how it relates to the enriched calcites spatially. The data in Figure 9 plots either within multiple zones, or at the borders of these empirically defined zones on the plots, which is not strongly convincing of source.

Response: We have added a sentence on how calcites and gas data are related: “The sampling site of the gas data in borehole VM2 corresponds to the uppermost ^{13}C -rich calcites in this borehole, whereas the other gas-sampled borehole (VM5) is just adjacent to other boreholes sampled for calcite (VM- and 01-boreholes).” at lines 250-253. We have also added a statement that it is indeed difficult to interpret source based on the fact that the gas data plots in overlapping or on borders between fields in Fig 8a and b. This was added after the discussion of the hydrogen isotopes, as both of a and b plots on the borders of fields. “Overall, the position of the gas samples within multiple zones, or at the borders of the empirically defined zones on the discrimination plots in Figure 8a and b shows that these plots alone are not diagnostic for any single process and/or gas origin.” At lines 288-291. Later in the text when $\delta^{13}\text{C}_{\text{CO}_2}$ is discussed, it is more obvious that microbial gas is important. Even more so when we discuss the ^{13}C -rich calcite we can confirm again that microbial methanogenesis has been widespread (see below), as also is accepted by the reviewer two comments below this one.

Line 267 The argument regarding the ethane and propane is plausible, however it does not address the lack of evidence of degradation of the butane in the same sample. Further this is only one sample from a complex system. The authors would be able to make much more effective arguments with access to further data if that can be obtained from the industrial owners of the data.

Response: Good point regarding butane, which deserves further clarification. We have added: “The lack of ^{13}C -enrichment in C_4 compared to C_3 can be due to that propane is biodegraded most rapidly of the hydrocarbons, already at slight levels of biodegradation⁴⁸. *n*-alkanes are preferentially utilized during biodegradation of oil and gas⁴⁶.” At lines 302-304. There is high *i*- C_4 /*n*- C_4 ratios in these samples (“VM5” are two samples, 1 and 2, with 3 and 1 subsamples, respectively). The high *i*- C_4 /*n*- C_4 ratios are discussed in a later section already, and we have expanded that section on degradation (see below).

Line 300 I agree microbial methane appears to be present

Response: Yes, the ^{13}C -enrichment is strong and widespread.

Line 307 I am not convinced that degraded hydrocarbons are the source of the DIC for methane production

Response: There is plenty of support for biodegradation in the system and we have added much more discussion (as detailed above). The reviewer is right that this may not be the only source for DIC and we have discussed the suggestion of the reviewer that DIC from limestone is of importance and added our explanation and clarification to a comment above.

Line 382 I agree that there is evidence of sulfur cycling within the system, however not that it can be related to production of the CO₂. I can accept that there may be evidence of anaerobic methane oxidation generating the observed $\delta^{13}\text{C}$ depleted carbonates.

Response: The pyrite data provide support for microbial sulfate reduction, together with anaerobic methane oxidation, which is a common syntrophic relationship (ANME and SRB). Sulfate reduction in relation to methanogenesis is probably very limited, as the reviewer points out. We have now, following the reviewers remark, changed focus to more plausible degradation pathways of the organics, mainly fermentation, and that (especially in the limestone aquifer, DIC can originate from other sources, i.e. wall rock) and that H₂ coming from fermentation of organic matter in bitumen and seep oil may be important substrate for the microbial communities to perform methanogenesis. See inserted text to other remarks above to see what changes have been made to the manuscript.

Line 426 I am not convinced of the conceptual model presented. I think the potential sources of DIC related to the carbonates need to be dealt with. I do not see strong evidence of biodegradation of the organics. But I do agree there appears to be evidence of microbial methanogenesis and methanotrophy recorded in the carbonates.

Response: As detailed above, we have opened up for more complex pathways and other sources of DIC than organic matter oxidation by MSR (that is no longer considered to be a major source) . We thank the reviewer for pointing this out.

Line 477 I am not convinced that the microbial methanogenesis has produced large quantities of methane

Response: We are not claiming that there are large quantities of methane produced with this process globally. We are presenting a case from Siljan and open up for more investigation of the vast upper crust (the sediment-igneous rock transition). There are very high gas pressures locally at Siljan, but these may represent to more or less sealed off isolated pockets in the

crater fracture system where methane can be accumulated over time. This study is focused to detection of long-term microbial processes in the impact crater and suggests that the geometry of the impact structure with deep penetration of joints that connects the crystalline bedrock aquifer with a black shale bearing sedimentary strata above gives ideal conditions for long-term methanogenesis of the deep biosphere communities. Gas volume estimates will be the focus of future work by the prospectors (test production wells etc) and require more studies from other sites to assess the global significance.

- Will the paper be of interest to others in the field?

This is an interesting system to address, however, the authors need to be more convincing of the explanation of what is happening and the breadth of its implications

Response: See answer to this comment above in the general assessment.

- Will the paper influence thinking in the field?

I am not sure how much this will influence thinking in the field. It is an interesting system, but I am not convinced it is highly unique by the data and arguments presented.

Response: This is the most extensive mineral-gas-geochronology study of an impact structure yet reported. It is the first that combines dating of minerals formed following microbial activity and combines it with gas data from the same boreholes. It is the first thorough study of long-term methane formation and consumption in a large impact structure which can have implications for understanding the deep unexplored microbial communities of such systems. We hope the reviewer can find the updated model and implications satisfactory.

- Are the claims convincing? If not, what further evidence is needed?

See comments above

Response: See answers above.

- Are there other experiments that would strengthen the paper further? How much would they improve it, and how difficult are they likely to be?

Further presentation of water and gas chemistry and isotopes would potentially fill out the argument. But I recognize that this data may not be easily obtained.

Response: We have gathered some more data about the gas (+water/gas sample), discussed in detail previous studies of seep oil and bitumen, as well as giving more detailed interpretations of these and on biomarker data (see answer above about biodegradation). We have now also incorporated and discussed water chemistry data from the aquifers (from adjacent boreholes in the granite aquifer of the crater, involving concentrations of HCO₃, SO₄, Cl).

Beside this, there are basically no other data available from the industrial owner or from previous studies.

- Are the claims appropriately discussed in the context of previous literature?
Yes

- If the manuscript is unacceptable in its present form, does the study seem sufficiently promising that the authors should be encouraged to consider a resubmission in the future?

I was not convinced that the authors have demonstrated the novelty and significance of the story sufficiently.

- Is the manuscript clearly written? If not, how could it be made more accessible?

The manuscript is well written

- Could the manuscript be shortened to aid communication of the most important findings?

There are some points, like the analysis of the organic residue by SIMS, that do not clearly currently contribute to the story.

Response: We have deleted this part.

- Have the authors done themselves justice without overselling their claims?
The claims of the paper need to be more clearly presented.

Response: We have made a complete overhaul of the methane formation pathway following the reviewer's advice and kept the claims that are based on the mineral signatures that the reviewer seems to approve.

- Have they been fair in their treatment of previous literature?
Yes

- Have they provided sufficient methodological detail that the experiments could be reproduced?

Yes

- Is the statistical analysis of the data sound?

Not a key component

-

- Should the authors be asked to provide further data or methodological information to help others replicate their work? (Such data might include source code for modelling studies, detailed protocols or mathematical derivations).

It was difficult to find some of the supporting material that was located in the excel file. Some of this data might warrant inclusion in the paper.

Response: We have included the CO₂ concentrations as requested above.

- Are there any special ethical concerns arising from the use of animals or human subjects?

No

Reviewers' comments:

Reviewer #1 (Remarks to the Author):

The authors have done a thorough job when the revising the manuscript.

Reviewer #2 (Remarks to the Author):

I found this version of the manuscript much more effective than the first version. The authors made significant efforts to respond to my queries and the minor points raised by the other review, as the authors note, I think it significantly improved the manuscript. There are still some points that I think the authors need to address in order to clarify the current version. Two main points, noted below, that need clarification are 1) the extent to which abiotic contributions to the methane pools can be ruled out and 2) the discussion of phospholipids in relationship to methanogenesis. With respect to point one, the author's note that abiotic contributions cannot be ruled out, but explore this possibility less than the other options and subsequently discuss the sources as if these contributions are not present. They must either actually find a way to rule them out or incorporate their potential presence in their discussion. Given that this site was one where the idea of abiotic methane contributions was proposed, this perspective requires significant attention. It will not really affect their argument for microbial methanogenesis, but has implications for the arguments regarding the role of biological respiration of the organics as a source of carbon. With respect to the second point, I assume the authors are aware that methanogens are archaea and do not produce phospholipids. Thus, while they show evidence of microbial activity, it would be only bacterial or eukaryotic. It would not support methanogenesis, but would support heterotrophy, fermentation or methanotrophy. Overall I think this study contributes to our understanding of signatures of microbial activity in subsurface systems. However, I am not certain that it extends our perspective to a great degree. This study is interesting and, if the authors address these issues and the other comments below, I think this manuscript would be publishable. I am not certain that it will strongly influence thinking in the field. I think it would be highly appropriate for a leading geochemical journal such as GCA. But I am not sure it will be of broad interest beyond the geochemical community.

Specific comments

Line 129 How do you sandwich sedimentary rocks between granite? This may have been discussed in other papers, but is relevant both to understanding the site interpretation and the presence of absence of biological activity.

Line 159 The depth in parentheses is not optimal, suggest re-wording it

Line 246 Suggest adding the comment that this predates the impact. Not that the reader should not remember, but it emphasizes the post impact nature of the younger calcites which the authors are focusing on.

Line 273 Bacterial ethanogenesis was also demonstrated near gas wells in western Canada by Taylor, Sherwood Lollar, Wassenaar Environ. Sci. Technol. 2000, 34, 4727-4732 and should be cited.

Line 284 However, the authors are not convincingly able to discount abiotic sources. The isotope ranges for these processes are empirical, so being just outside it is not entirely convincing. And the evidence of mixing of thermogenic and microbial could also support an abiotic origin. The authors recognize the process, but cannot discount it as strongly as they do based on the data alone. As this has implications to later discussion it can't be discarded without evidence.

Line 287 H isotope ranges are also known for abiotic sources, what is the outcome of this comparison?

Line 295 This statement should be clarified. What precursors? What pathway?

Line 315 Clarify why this lack of isolation means it cannot be ruled out. I agree abiogenic sources cannot be ruled out.

Line 325 I agree with the authors on the mixed nature, and potentially biodegraded nature of the gases. But it is not clear that abiotic contributions can be ruled out. The authors do say this, but

their discussion implies this is the least likely scenario. Given the historical proposal of abiotic methane at the site I think they should justify this more strongly. I don't think it will negate their overall story concerning microbial inputs.

Line 333 – I still am okay with the enriched carbonate statement. But I presume the authors are aware that methanogens are archaea and do not produce phospholipids nor the same fatty acids as bacteria. So while the presence of fatty acids can be argued to indicate microbial presence, they should not imply that it supports the presence of methanogens. It does however support the presence of fermenters or sulphate reducers.

Line 352 The grammar of this statement is difficult to follow, please clarify

Line 382 This section is much better

Line 393 The use of H₂ by methanogens should reduce H₂ and increase CH₄ so the correlation should be inverse. A concurrent increase would argue against methanogenesis by CO₂ reduction.

Line 407 The definition of secondary methane is good, but it had not been given when the term was first used above. It should be defined at first use (I believe line 195)

Line 415 But abiotic sources may also exist and need to be discussed.

Line 446 Again this section is much better than in the previous version

Line 480 Abiotic contributions were not ruled out so should be discussed.

Line 526 Which environment are the authors referring to? Impact structures? If so, they are surely not the largest of any environment on Earth.

Reviewer #1 (Remarks to the Author):

The authors have done a thorough job when the revising the manuscript.

Response: We thank the reviewer for constructive remarks on the first version of the manuscript and focus the revision to the points raised by reviewer#2 below.

Reviewer #2 (Remarks to the Author):

I found this version of the manuscript much more effective than the first version. The authors made significant efforts to respond to my queries and the minor points raised by the other review, as the authors note, I think it significantly improved the manuscript. There are still some points that I think the authors need to address in order to clarify the current version. Two main points, noted below, that need clarification are 1) the extent to which abiotic contributions to the methane pools can be ruled out and 2) the discussion of phospholipids in relationship to methanogenesis. With respect to point one, the author's note that abiotic contributions cannot be ruled out, but explore this possibility less than the other options and subsequently discuss the sources as if these contributions are not present. They must either actually find a way to rule them out or incorporate their potential presence in their discussion. Given that this site was one where the idea of abiotic methane contributions was proposed, this perspective requires significant attention. It will not really affect their argument for microbial methanogenesis, but has implications for the arguments regarding the role of biological respiration of the organics as a source of carbon. With respect to the second point, I assume the authors are aware that methanogens are archaea and do not produce phospholipids. Thus, while they show evidence of microbial activity, it would be only bacterial or eukaryotic. It would not support methanogenesis, but would support heterotrophy, fermentation or methanotrophy. Overall I think this study contributes to our understanding of signatures of microbial activity in subsurface systems. However, I am not certain that it extends our perspective to a great degree. This study is interesting and, if the authors address these issues and the other comments below, I think this manuscript would be publishable. I am not certain that it will strongly influence thinking in the field. I think it would be highly appropriate for a leading geochemical journal such as GCA. But I am not sure it will be of broad interest beyond the geochemical community.

Response: We are pleased to see that the reviewer finds this version much more effective than the previous version. We agree with the two general comments, that potential abiotic gas contribution should be discussed in more detail (especially for the granite aquifer), and that it should be more emphasized that phospholipids are microbial remnants originating from e.g. fermentation and sulfate reduction processes rather than archaeal processes. Consequently, we have added discussion about these two matters, as detailed below in the line-by-line remarks/responses. Once again, we thank the reviewer for constructive comments that certainly has helped to improve the manuscript.

Specific comments

Line 129 How do you sandwich sedimentary rocks between granite? This may have been discussed in other papers, but is relevant both to understanding the site interpretation and the presence of absence of biological activity.

Response: More information has been added: “At Stumnsnäs, large scale impact-related faulting has caused a slab of Proterozoic granite to overthrust the sedimentary successions which thus are sandwiched in between blocks of granite at 196-286 m depth^{33,34}. Oil stains and calcite mineralizations have been observed in limestone fractures in the sedimentary successions at Stumnsnäs^{33,34}.”

Line 159 The depth in parentheses is not optimal, suggest re-wording it

Response: Re-phrased: “¹³C-rich calcite occurs up to 212 m above the sediment-granite contact...”

Line 246 Suggest adding the comment that this predates the impact. Not that the reader should not remember, but it emphasizes the post impact nature of the younger calcites which the authors are focusing on.

Response: Good suggestion, we re-phrased: “The older type predates the impact and dates back to 600-400 Ma as indicated by the U-Pb dating of two samples...”

Line 273 Bacterial ethanogenesis was also demonstrated near gas wells in western Canada by Taylor, Sherwood Lollar, Wassenaar Environ. Sci. Technol. 2000, 34, 4727-4732 and should be cited.

Response: Good suggested addition, we added: “Regarding the higher hydrocarbons, it has been demonstrated that microbial ethano- and propanogenesis occur in deeply buried marine sediments⁴², and the former also near gas wells in western Canada⁴³.” and added the requested citation.

Line 284 However, the authors are not convincingly able to discount abiotic sources. The isotope ranges for these processes are empirical, so being just outside it is not entirely convincing. And the evidence of mixing of thermogenic and microbial could also support an abiotic origin. The authors recognize the process, but cannot discount it as strongly as they do based on the data alone. As this has implications to later discussion it can't be discarded without evidence.

Response: We have added more discussion about potential abiotic contribution by discussing diagnostic gas composition signatures that was not done in detail before (e.g. that the relatively heavy $\delta^{13}\text{C}_{\text{C}_2}$ and $\delta^{13}\text{C}_{\text{C}_3}$ (addition: “These relatively heavy $\delta^{13}\text{C}_{\text{C}_2}$ values and even heavier $\delta^{13}\text{C}_{\text{C}_3}$ values speak against a significant contribution from abiotic gas, which generally features decreasing $\delta^{13}\text{C}$ values with higher carbon number of the homologues⁴⁴.”). In addition, we have, here and elsewhere, concluded that an abiotic mixing end-member cannot be ruled out, accordingly “Taken together, these data suggest that the gas is to a large extent microbial, to a significant extent

thermogenic, and possibly to a minor extent abiotic.”. This means that we have now presented more support for that abiotic contribution likely is very minor (at least in the sediments), yet added the potential of such contribution when mixing and gas sources are discussed, as it cannot be ruled out completely.

Line 287 H isotope ranges are also known for abiotic sources, what is the outcome of this comparison?

Response: Here we have also added more details for comparison with abiotic sources (comparing with empirical cited data from other sites, in the discrimination diagram) to give more input to that discussion. Addition made: “Additionally, the Siljan gas samples are overlapping with H isotope ranges of abiotic sources at other sites¹⁰, which means that abiotic contribution cannot be ruled out based on the H isotope composition alone.”

Line 295 This statement should be clarified. What precursors? What pathway?

Response: We added some more information and incorporated the comment for line 407 below. Added/updated: “Methane formed following secondary microbial utilization of primary thermogenic hydrocarbons (e.g. petroleum, seep oils and lighter hydrocarbons) is commonly termed secondary methane. The heavy $\delta^{13}\text{C}_{\text{CO}_2}$ values detected are typical for secondary microbial methane⁴⁶, which is supported by other biodegradation signatures.”

Line 315 Clarify why this lack of isolation means it cannot be ruled out. I agree abiogenic sources cannot be ruled out.

Response: We have added more information on this matter (for the granitic rock part). “Although abiotic gas contribution cannot be directly identified in the investigated (dominantly sedimentary) aquifer, it cannot be ruled out, at least not in the deep granite fracture system, because none of the borehole samples isolates gas from the crystalline aquifer alone. There is thus a theoretical possibility that abiotic gas from the granite fractures is masked, to an unknown degree, in the analysed gas samples by gases from the sedimentary rock fractures.”

Line 325 I agree with the authors on the mixed nature, and potentially biodegraded nature of the gases. But it is not clear that abiotic contributions can be ruled out. The authors do say this, but their discussion implies this is the least likely scenario. Given the historical proposal of abiotic methane at the site I think they should justify this more strongly. I don't think it will negate their overall story concerning microbial inputs.

Response: We agree that abiotic contribution cannot be fully ruled out, at least not in the deeper granite system. However, we have added some more discussion that speak against abiotic contribution in the sedimentary aquifer (above). We have here added a concluding statement to this section, so it is clearer what our main interpretation is, and that we are aware of the potential abiotic contributor. “This is in accordance with the mixed gas of interpreted dominantly microbial and thermogenic origin detected in the

present study (Fig. 8). Hence, although the results presented here and previously are not generally supportive of abiotic gas, such gas cannot be fully ruled out, at least not in the deeper, granitic system.”

Line 333 – I still am okay with the enriched carbonate statement. But I presume the authors are aware that methanogens are archaea and do not produce phospholipids nor the same fatty acids as bacteria. So while the presence of fatty acids can be argued to indicate microbial presence, they should not imply that it supports the presence of methanogens. It does however support the presence of fermenters or sulphate reducers.

Response: This is generally a correct remark. Although we have not used the biomarkers as definite proof for presence of archaea (methanogens), rather as overall proof for microbial activity *in situ*, we have clarified this discussion by adding: “The detected FA can obviously be tied to fermentation⁵³ and/or sulfate reduction by bacteria⁵⁴, in line with the S isotope record in pyrite in the fractures. Even though methanogenesis is commonly attributed to archaea, which do not produce phospholipid fatty acids, recent studies highlight bacterial methane production through N-fixation together with CO₂⁵⁵, which also is a plausible process in the fractures. Furthermore, soil-derived fungi are able to produce high amounts of methane through biodegradation in relationship with methanogens⁵⁶, but also without⁵⁷, and high diversity of fungi have been detected in the continental crust^{58,59}. Fungi produce phospholipids and other FA than bacteria, but potential presence of fungi at Siljan is yet unexplored. However, the particular FA detected cannot be used as diagnostic markers for methanogens, in contrast to the heavy $\delta^{13}\text{C}_{\text{calcite}}$ -values.”

Line 352 The grammar of this statement is difficult to follow, please clarify

Response: We changed and split up into two sentences accordingly: “Both the sedimentary rocks and the deeper granite fractures contain widespread bitumen and seep oil stains. However, the lack of other carbon sources in the granite fractures lead to higher degree of microbial utilization of bitumen and oil stains in the granite fractures (Fig. 7).”

Line 382 This section is much better

Response: We are very satisfied to hear that, as we put a lot of effort into improving this section. We agree and thank the reviewer for the constructive comments on previous version of this section.

Line 393 The use of H₂ by methanogens should reduce H₂ and increase CH₄ so the correlation should be inverse. A concurrent increase would argue against methanogenesis by CO₂ reduction.

Response: Correct remark. We have deleted this sentence as the values from different boreholes gave rather mixed interpretations and gas compositions are very low in the old investigations.

Line 407 The definition of secondary methane is good, but it had not been given when the term was first used above. It should be defined at first use (I believe line 195)

Response: See answer to comment about line 295.

Line 415 But abiotic sources may also exist and need to be discussed.

Response: We added a sentence about this, and that it is in particular of interest in the granite system: “In the deeper granite system, contribution from abiotic gas sources may also have been involved in the biodegradation feedstock.”

Line 446 Again this section is much better than in the previous version

Response: We agree and once again thank the reviewer for the constructive comments on previous version of the section.

Line 480 Abiotic contributions were not ruled out so should be discussed.

Response: We added “and perhaps a minor abiotic gas fraction” to show that we have not ruled out this possibility.

Line 526 Which environment are the authors referring to? Impact structures? If so, they are surely not the largest of any environment on Earth.

Response: This particular statement refers to the upper crystalline crust in general. We have specified this in the revised manuscript.